# Prefrontal–bed nucleus of the stria terminalis physiological and neuropsychological biomarkers predict therapeutic outcomes in depression

Linbin Wang [1,2,3,4], Yingying Zhang [2,4], Yuhan Wang[1,4], Qiong Ding [3], Luling Dai[1], Kejia Hu[1], Kuanghao Ye[1], Xin lv [1], Xiaoxiao Zhang[1], Alekhya Mandali [3], Luis Manssuer[3], Saurabh Sonkusare [3], Yijie Zhao [2], Peng Huang[1], Xian Qiu[1], Yixin Pan[1], Yijie Lai[1], Dianyou Li [1], Wei Liu[1], Shikun Zhan[1], Bomin Sun [1] ✉ & Valerie Voon [1,2,3] ✉

Therapeutic options for refractory depression are urgently needed. We conducted a deep brain stimulation (DBS) randomized controlled trial of the bed nucleus of the stria terminalis (BNST), an extended amygdala structure, and nucleus accumbens (NAc) in 26 refractory depression patients to assess treatment efficacy and predictors of response. BNST-NAc DBS had a 50% depression response rate and 35% remission rate in the open-label phase. We identified an objective intracranial physiological biomarker using acute and chronic intracranial recordings, machine learning, and an integrated framework combining electrophysiology, neuroimaging, and behavior: lower BNST theta and prefrontal-BNST coherence with top-down connectivity predicted better depression outcomes and quality-of-life after chronic stimulation at 3, 6 and 12 months, confirmed across eyes -open and -closed states and machine learning. We identified a physiology-guided connectivity network involving dorsal anterior cingulate and lateral inferior frontal cortex tracts. These biomarkers, linked to negative emotional bias and anxiety, highlight the efficacy of BNST-NAc DBS for refractory depression and has potential broader clinical implications. ClinicalTrials.gov registration: NCT04530942.

Major depressive disorder (MDD) is one of the most common disabling global mental health problems with high rates of treatment resistance[1]. Therapeutic options for refractory depression are urgently needed. Deep brain stimulation (DBS) is a neurosurgical procedure involving delivery of high-frequency stimulation to deep brain structures to modify pathological activity with long-term established efficacy in Parkinson's disease and movement disorders[2] and obsessive-compulsive disorder (OCD)[3].

DBS targeting the medial prefrontal cortex (mPFC)-limbic network associated with impaired emotion and reward processing in MDD, have demonstrated potential therapeutic efficacy[4–7]. While open-label and longitudinal studies have shown promise since the first

[1]Department of Neurosurgery, Center for Functional Neurosurgery, Ruijin Hospital, Shanghai Jiao Tong University School of Medicine, Shanghai, China. [2]Institute of Science and Technology for Brain-Inspired Intelligence (ISTBI), Fudan University, Shanghai, China. [3]Department of Psychiatry, University of Cambridge, Cambridge, UK. [4]These authors contributed equally: Linbin Wang, Yingying Zhang, Yuhan Wang. ✉e-mail: bomin_sun@163.com; vv247@cam.ac.uk

report on subcallosal cingulate cortex (SCC) DBS[8,9], randomized controlled trials (RCT) for depression have had mixed results with variability related to issues including targeting precision in the context of individual variability in relevant white matter tracks, study design to allow individual optimization, and the heterogeneity of depression[5–7]. In addition to the SCC, which targets the intersection of three limbic-relevant white matter tracts, two major randomized trials targeting the nucleus accumbens/ventral internal capsule have shown mixed findings in depression[5–7]. Other targets highlighted the mesolimbic dopaminergic system with a small positive randomized trial of the medial forebrain bundle and small pilot trials of the lateral habenula[10–15]. The absence of objective methods to assess interoceptive subjective depression-related symptoms further complicates treatment evaluation. The variability in treatment outcomes in the randomized trials hinders the establishment of DBS as a standard treatment for depression despite its urgent demand, highlighting the importance of identifying reliable biomarkers of treatment response towards personalized and precision therapeutics.

The bed nucleus of the stria terminalis (BNST), part of the extended amygdala, is implicated in fear processing, assigning valence and social interactions[16–22], which are important constructs underlying depression. The anterior BNST overlaps with the posterior nucleus accumbens with current spread from nucleus accumbens/ventral capsule targets possibly influencing BNST. The BNST has also shown preliminary efficacy as a DBS target for MDD[23–25] and OCD[26] and has been suggested to be relevant to improvements in depression[27]. This includes a pilot open-label study suggesting potential efficacy, although a follow-up small RCT (*n* = 8) did not show efficacy, likely due to the small sample size, and lack of individualized optimization of stimulation parameters with an initial fixed cross-over RCT of contacts at low and moderate intensity and stimulation of the lower two and top two contacts[24,28]. The BNST is well-placed to play an integrative role in emotional processing and regulation with extensive connections with hypothalamic, autonomic, and midbrain neurotransmitter structures[17].

Here, we studied a relatively rare group of refractory MDD patients who underwent DBS using dual monopolar stimulation of an 8-contact electrode allowing multitargeting and stimulation of the BNST, nucleus accumbens (NAc), and ventral internal capsule (VIC). We leveraged the capacity to record from intracranial physiology with high spatial precision and signal-to-noise ratio. Our primary aim was to derive subtype-sensitive objective biomarkers that predicted treatment efficacy of chronic BNST-NAc DBS. We then confirmed with machine learning and validated across eyes-open and eyes-closed states (Fig. 1a). Utilizing wireless intracranial electrophysiology recording, we extended evidence of the identified BNST dynamics tracking both stimulation effects of BNST-NAc DBS and transient individual mood fluctuations (Fig. 1b). Within an integrated multi-dimensional predictive framework, we bridged electrophysiology, neuroimaging and behavior, to interrogate relevant structural and functional connectivity, patterns of network interaction, and potential psychological correlates.

## Results
### Patient characteristics and study design
Between July 2021 and April 2024, we conducted a prospective, double-blinded crossover RCT (ClinicalTrials.gov identifier: NCT04530942) at Ruijin Hospital, Shanghai Jiaotong University School of Medicine (Shanghai, China) (RCT results submitted elsewhere). A total of 524 patients were screened for eligibility, of which 26 were included in the study. Patient demographic characteristics (88% male; mean age 31.7 years (SD 9.9); mean age of depression onset 18.9 years (SD 7.1) and mean hospitalization 2.5 times (SD 0.5)) are further summarized in Supplementary Table 1.

The study started with a longitudinal, open-label trial lasting at least 6 months followed by a double-blind, 4-week randomized

crossover with 2 weeks of active versus 2 weeks of sham stimulation. We obtained data from both acute externalized and chronic wireless electrophysiology to establish predictive biomarkers of treatment response. Subjects were externalized following the initial implantation. During this perioperative period, we acquired 5-min resting state recordings of local field potential (LFP) and frontal scalp electroencephalogram (EEG) data, both with eyes closed and open. Seventeen patients (eyes-closed dataset) were included following data inspection and confirmation of target localization using Lead-DBS with cross-validation in the eyes-open dataset (*n* = 15) to exclude state bias[29]. Postoperatively, we streamed multiple 5-min rest episodes of intracranial electrophysiological data labeled with momentary assessment of emotional states from 10 patients (8–17 episodes each) over the crossover period. All postoperative recordings were performed in-clinic. A behavioral affective task was performed at the perioperative period to capture subjective emotional bias ratings.

Throughout the study, patients were assessed at baseline (1–3 days pre-surgery) and then monthly post-surgery using standardized measures for depression (Hamilton Depression Scale-17 [HAMD-17], Montgomery-Asberg Depression Rating Scale [MADRS]), anxiety (Hamilton Anxiety Scale-14 [HAMA-14]), anhedonia (Dimensional Anhedonia Rating Scale [DARS]), quality-of-life (World Health Organization Quality of Life-BREF [WHOQOL-BREF], 36-Item Short Form Survey Instrument [SF-36]), and disability (Sheehan Disability Scale [SDS]).

### Clinical outcomes: phenotypic divergence in response to BNST-NAc DBS
During the open-label phase, patients were assessed monthly clinically for symptom severity, with adjustments made to their parameters approximately every two weeks as needed. At the last follow-up (19 ± 8.5 months), all patients received multi-site monopolar stimulation, customized through programming to meet individual needs. The volume of tissue activated predominantly overlapped the BNST, extending to include the NAc and VIC (Supplementary Fig. 1a and Supplementary Table 2). Stimulation parameters are summarized in Supplementary Table 3. Based on the HAMD scores at last follow-up in the open label phase, 13 of 26 patients (50%) were classified as responders, with 9 of them (35%) achieving remission, while the remaining 13 patients (50%) were classified as non-responders, with 7 of them (27%) being partial responders. Stimulation resulted in significant reductions in HAMD scores by an average of 10.1 points (SD 7.1, 95% CI 7.2–13.0, $t(25) = 7.26$, FDR adjusted $p < 0.001$), MADRS scores by 13.5 points (SD 7.8, 95% CI 10.4–16.6, $t(25) = 8.86$, FDR adjusted $p < 0.001$), and HAMA scores by 10.4 points (SD 7.2, 95% CI 7.5–13.3, $t(25) = 7.33$, FDR adjusted $p < 0.001$). Although DARS scores increased, the significance did not hold after correction for multiple comparisons. Additionally, quality-of-life measures (SF-36, WHO-BREF) and disability scores (SDS) showed significant improvement (Supplementary Fig. 2a and Supplementary Table 4).

At preoperative baseline, anxiety/depression and anhedonia were dissociated at baseline with HAMD, MADRS, and HAMA highly correlated (all $p < 0.001$ and survived after FDR correction) but not with DARS ($p > 0.05$), indicating potentially differential clinical phenotypes. While depression, anxiet,y and anhedonia were correlated in the short-term postoperatively at 3-month, only depression and anxiety were correlated at 6-month and 1 year (all $p < 0.01$ and survived after FDR correction, correlation coefficients of Spearman's r summarized in Supplementary Fig. 2b).

### Predictive biomarker: BNST theta biomarkers predict treatment response
We then focused on intracranial electrophysiological biomarkers within the prefrontal-limbic circuitry assessing prediction of

## a. Integrated Neuropsychopathological Predictive Framework

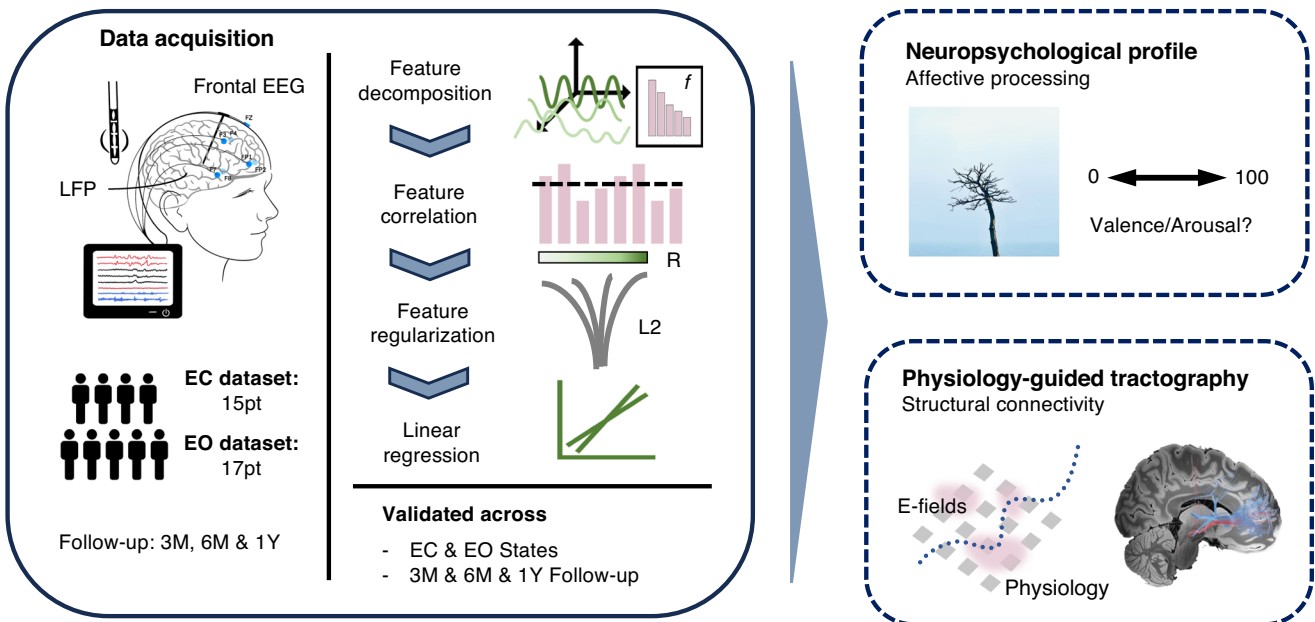

## b. Longitudinal Dynamic Tracking

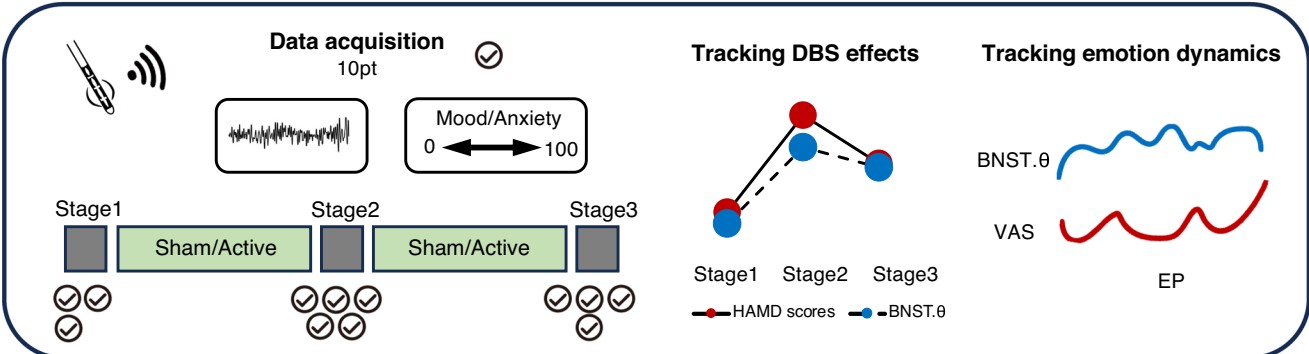

**Fig. 1 | Study overview. a** An integrated neuropsychopathological framework for identification of electrophysiological biomarkers predicting BNST-DBS efficacy. Resting-state LFP and frontal EEG were recorded perioperatively with externalized DBS electrodes. Seventeen patients (eyes-closed dataset) were included, with cross-validation in the eyes-open dataset (*n* = 15). Spectral features were extracted, correlated with 3-, 6-, and 12-month clinical outcomes, and entered into a penalized ridge regression model with permutation-based feature selection. Perioperative affective tasks captured emotional bias ratings, and prefrontal–BNST networks were mapped with physiology-guided tractography. **b** Longitudinal clinical and physiological tracking. Repeated 5-min resting-state recordings with momentary emotional ratings were obtained from 10 patients (8–17 sessions each) during the crossover period. DBS effects on BNST theta dynamics and their relationship to emotional state fluctuations were examined. EC eyes-closed dataset, EO eyes-open dataset, EEG electroencephalography, 3M 3 months, 6M 6 months, 12M 12 months, LFP local field potential, L2 L2 (ridge) regularization, pt patients, EP episodes, VAS visual analog scale, HAMD Hamilton Depression Scale, DBS deep brain stimulation, BNST bed nucleus of the stria terminalis.

therapeutic outcomes. To achieve this, we identified spectral features and functional connectivity within the BNST, NAc, hypothalamus, and prefrontal cortex that correlated with improvements [(pre-post)/pre] in depression (HAMD, MADRS), anxiety (HAMA), and anhedonia (DARS) at 3 months (*n* = 17). Significant features (unadjusted *p* < 0.05) indicative of therapeutic effects included BNST theta, beta and gamma power, prefrontal theta power and prefrontal-BNST theta coherence (Fig. 2a). Hypothalamic and NAc features were unrelated to therapeutic effects. Notably, we found only predictive effects and no disease state-related effects associated with baseline symptom severity.

We then fit the spectral features of prefrontal EEG, BNST, and their connectivity into a machine learning algorithm using penalized ridge regression model with permutation feature selection. Critically, BNST theta power and prefrontal-BNST theta coherence were the two most informative features for the training model in the eyes-closed dataset (Fig. 2b). Either of them alone was sufficient to predict the 3-month antidepressant and anxiolytic effects of BNST-NAc DBS but not anhedonia, with lower theta power (HAMD: $R^2$ = 0.55, *p* < 0.001; MADRS: $R^2$ = 0.32, *p* = 0.017; HAMA: $R^2$ = 0.54, *p* < 0.001; DARS: $R^2$ = 0.15, *p* = 0.129) and coherence (HAMD: $R^2$ = 0.31, *p* = 0.021; MADRS: $R^2$ = 0.20, *p* = 0.069; HAMA: $R^2$ = 0.30, *p* = 0.023; DARS: $R^2$ = 0.04, *p* = 0.431) predicting better improvement (Fig. 2c). We also confirmed their long-term predictive value at 6-month (*n* = 17) and 1-year (*n* = 15) (Fig. 2c and Supplementary Fig. 3c).

Using the eyes-open data set, we verified the biomarker is robust across different recording states. Our findings successfully replicated the predictive value of BNST theta power but not coherence in the eyes-open dataset (Supplementary Fig. 3a and c).

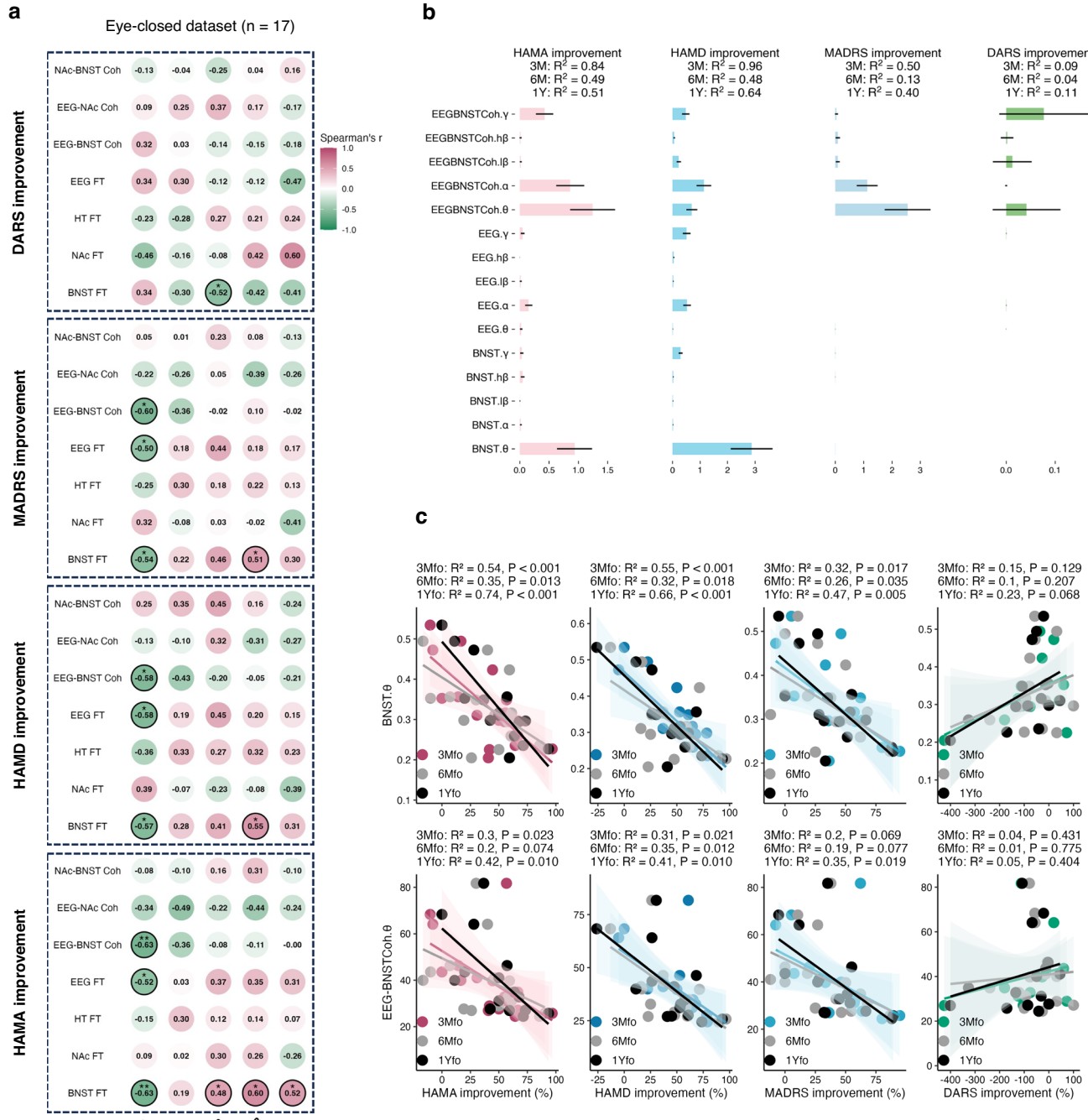

**Fig. 2 | BNST theta biomarkers predict treatment response. a** Correlation matrix between 3-month clinical improvements and spectral features at the perioperative baseline, black circle indicates unadjusted $p < 0.05$ for Spearman's correlation (two-sided). Source data are provided as a Source Data file. **b** Permutation feature importance of Ridge regression model. Error bar reflects SEM over 10,000 random iterations. **c** Linear regression of perioperative baseline BNST theta power and frontal-BNST theta coherence predicting 3-month ($n = 17$), 6-month ($n = 17$), and 1-year ($n = 15$) clinical improvements. Regression line shows the fitted mean and shading represents 95% CI. Annotations report $R^2$ and $P$ value for linear regression (two-sided). HAMD Hamilton Depression Scale, HAMA Hamilton Anxiety Scale, MADRS Montgomery-Asberg Depression Rating Scale, DARS Dimensional Anhedonia Rating Scale, BNST bed nucleus of the stria terminalis, FT Fourier Transform, Coh coherence, 3M 3 months, 6M 6 months, 12M 12 months, fo follow-up.

## Theta biomarkers involve a top-down prefrontal-BNST network

We investigated the direction of functional connectivity between prefrontal and BNST using frequency-resolved Granger prediction. Top-down theta band Granger causality at baseline correlated with theta power (prefrontal EEG: Spearman's $r = 0.51$, FDR adjusted $p = 0.04$; BNST: Spearman's $r = 0.72$, FDR adjusted $p = 0.006$) and coherence (Spearman's $r = 0.58$, FDR adjusted $p = 0.026$) and also correlated with depression and anxiety 3-month improvements (HAMA: Spearman's $r = -0.53$, $p = 0.028$; MADRS: Spearman's $r = -0.52$, $p = 0.031$; HAMD: Spearman's $r = -0.53$, $p = 0.027$; all survived after FDR correction). This was not observed for bottom-up theta band Granger causality ($p > 0.05$) (Fig. 3a and Supplementary Fig. 1b). Time reversal of the time series confirmed the directionality, with reversed correlations observed in bottom-up (HAMA: Spearman's $r = -0.28$, $p = 0.268$; MADRS: Spearman's $r = -0.57$, $p = 0.018$; HAMD: Spearman's $r = -0.48$, $p = 0.05$; uncorrected) but in top-down theta band Granger causality.

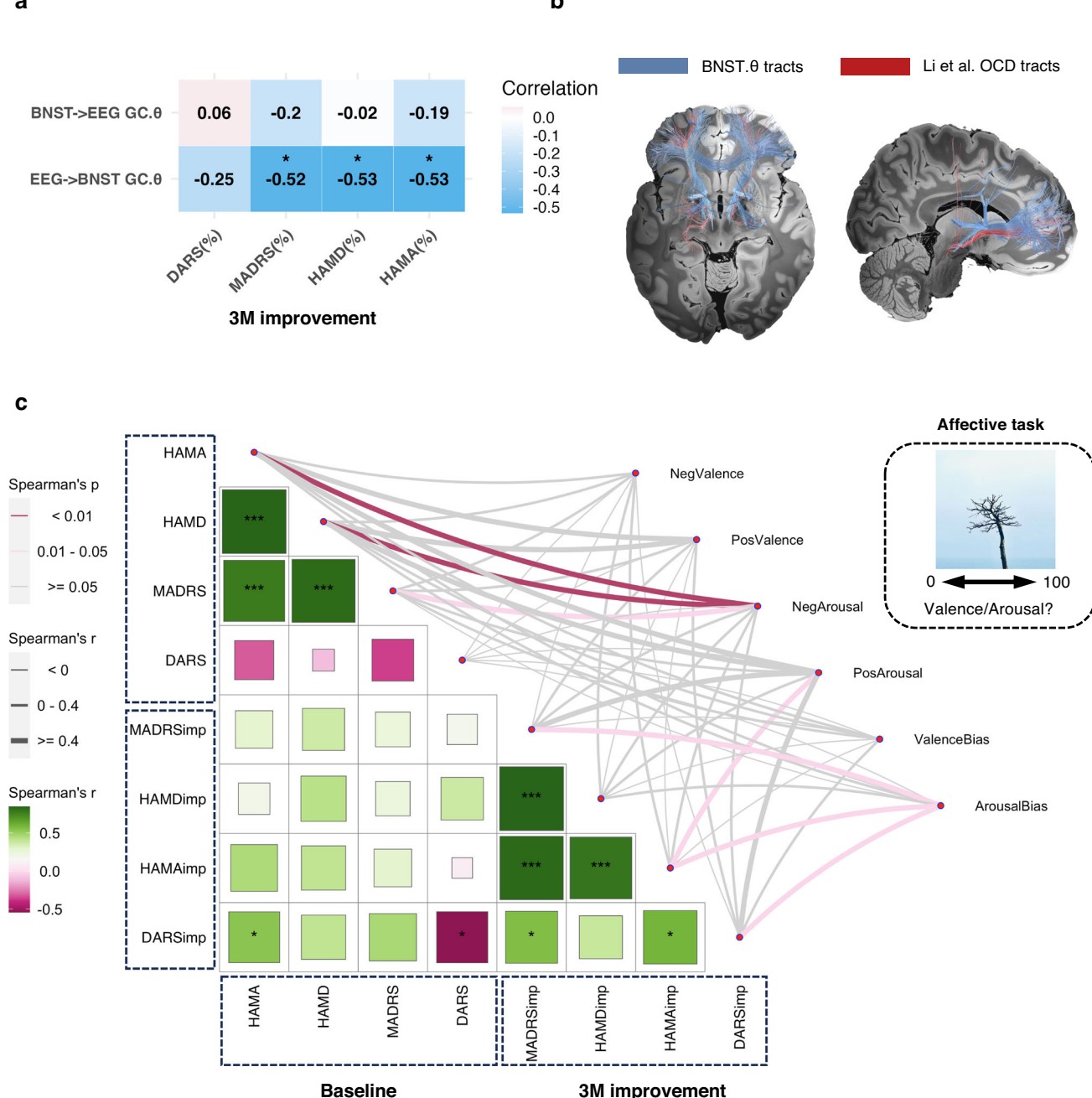

**Fig. 3 | Prefrontal–BNST theta biomarkers and neuropsychological predictors.** **a** Correlation matrix between theta band Granger causality and 3-month clinical improvements ($n = 17$). *FDR adjusted $p < 0.05$ for Spearman's correlation (two-sided). **b** Fiber tracts (blue) related to lower BNST theta power (predicting better clinical outcomes) overlaid with fiber tracts from the open-source tracts available for optimal OCD DBS targets (red). **c** Correlation map ($n = 17$) between symptom severity at baseline, 3-month treatment response and affective ratings (arousal and valence). *Unadjusted $p < 0.05$, **Unadjusted $p < 0.01$, ***Unadjusted $p < 0.001$ for Spearman's correlation (two-sided). HAMD Hamilton Depression Scale, HAMA Hamilton Anxiety Scale, MADRS Montgomery-Asberg Depression Rating Scale, DARS Dimensional Anhedonia Rating Scale, 3M 3 months, BNST bed nucleus of the stria terminalis, GC Granger Causality, NegValence negative valence, PosValence positive valence, NegArousal negative arousal, PosArousal positive arousal.

Next, we asked which prefrontal regions might be implicated in the prefrontal-BNST network based on our physiology-guided tractography analyses. Using fiber filtering in Lead-DBS[29], we stratified based on theta power and showed that lower theta power and coherence correlated with fiber tracts (unadjusted $p < 0.05$) projecting between BNST and dorsal anterior cingulate and lateral frontal cortex seeding from clinical BNST stimulated contacts (Fig. 3b and Supplementary Fig. 1c). Furthermore, we identified overlapping fiber tracts associated with improved depression (HAMD) and anxiety (HAMA) outcomes.

Pathways linked to greater depression improvement showed substantial overlap with physiology-guided tractography, whereas tracts associated with anxiety improvement were relatively sparse and primarily localized to subcortical regions (Supplementary Fig. 5a, b).

### Right-hemisphere dominance of theta-band biomarkers
Hemispheric lateralization is consistently observed in emotion processing, with the left hemisphere linked to positive or approach-related stimuli and the right to negative or withdrawal-related

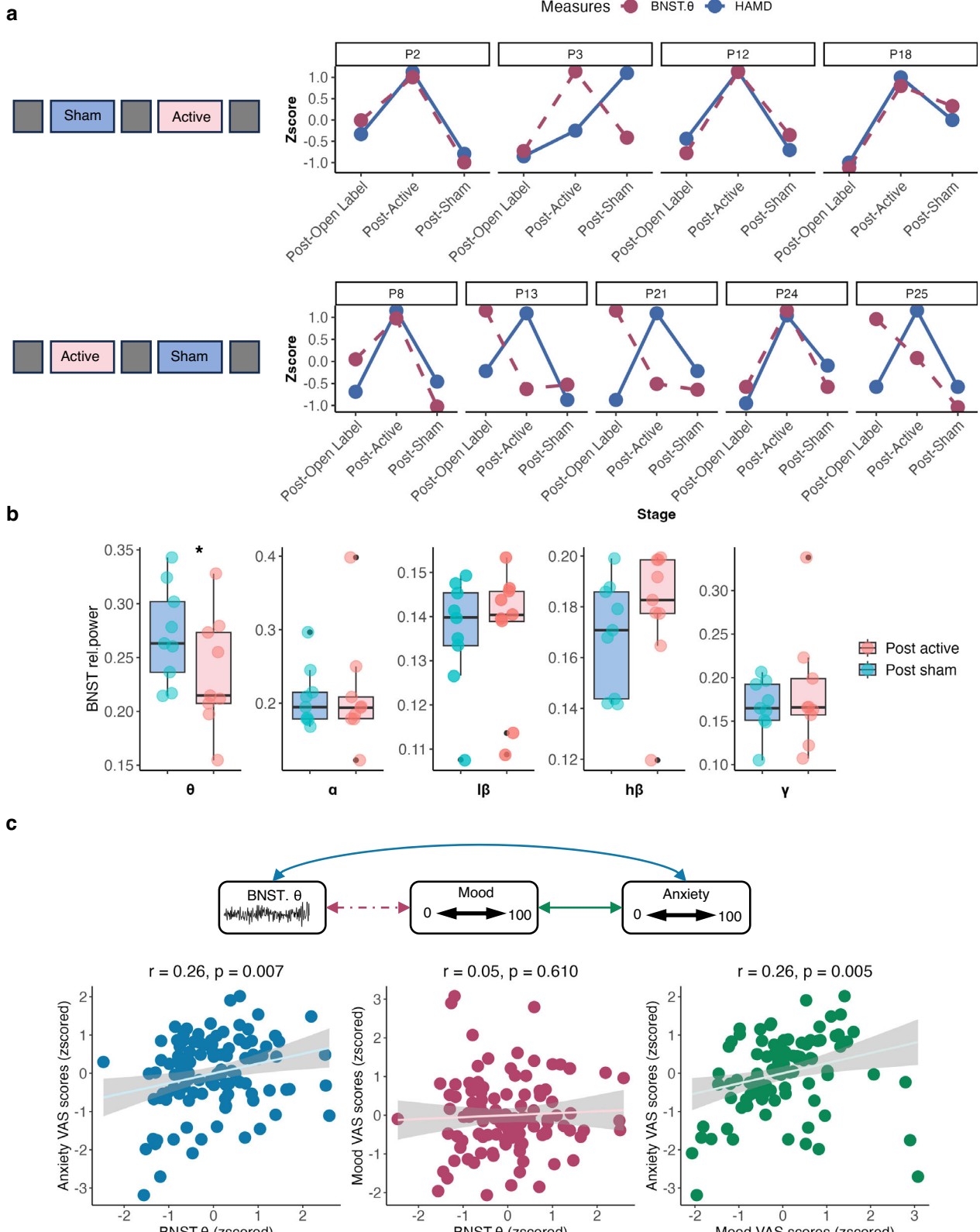

**Fig. 4 | BNST theta biomarker for longitudinal tracking. a** Simultaneous change of BNST theta power (red) and depression severity (HAMD, blue) in the randomized double-blind crossover phase. **b** BNST power across different frequency bands from post sham stimulation (2 weeks) stage to post active stimulation (2 weeks) stage ($n = 9$). Boxes show the interquartile range (25th–75th percentiles) with median as the center line. Whiskers extend to the most extreme data within 1.5 × IQR from the hinges; values beyond are plotted as grey points. *FDR adjusted $p = 0.04$ for Wilcoxon signed-rank sum test (two-sided). **c** Pearson's correlations between BNST theta power, mood VAS score, and anxiety VAS scores ($n = 110$). Regression line shows the fitted mean, and shading represents 95% CI (two-sided). BNST bed nucleus of the stria terminalis, VAS visual analog scale, rel.power relative power.

stimuli[30]. We assessed hemispheric differences in theta biomarkers by analyzing the left ($n = 13$) and right ($n = 14$) BNST separately and examining their correlations with 3-month clinical improvements. Notably, significant correlations with depression and anxiety improvement were observed in the right BNST (HAMA: Spearman's $r = -0.84$, $p < 0.001$; HAMD: Spearman's $r = -0.76$, $p = 0.002$; MADRS: Spearman's $r = -0.71$, $p = 0.004$; DARS: Spearman's $r = 0.47$, $p = 0.09$), while the left BNST showed no significant associations (HAMA: Spearman's $r = -0.29$, $p = 0.33$; HAMD: Spearman's $r = -0.15$, $p = 0.62$; MADRS: Spearman's $r = -0.33$, $p = 0.27$; DARS: Spearman's $r = 0.14$, $p = 0.64$).

To assess the anatomical specificity of prefrontal EEG channels and their alignment with tractography, we conducted an exploratory analysis between EEG measures (spectral power and EEG-BNST coherence) from each channel and improvements in depression (HAMD, MADRS), anxiety (HAMA), and anhedonia (DARS) at 3 months and controlled for multiple comparisons using FDR correction across 7 EEG channels. Notably, only theta power in FP2 (HAMA: Spearman's $r = -0.61$, FDR adjusted $p = 0.034$; MADRS: Spearman's $r = -0.59$, FDR adjusted $p = 0.046$) and F8 (HAMA: Spearman's $r = -0.66$, FDR adjusted $p = 0.026$; MADRS: Spearman's $r = -0.59$, FDR adjusted $p = 0.046$) showed significant associations with 3-month HAMA and MADRS improvements. Additionally, theta power across all prefrontal EEG channels was significantly correlated with 3-month HAMD improvements: FP1 (Spearman's $r = -0.59$, FDR adjusted $p = 0.025$), FP2 (Spearman's $r = -0.66$, FDR adjusted $p = 0.025$), Fz (Spearman's $r = -0.56$, FDR adjusted $p = 0.025$), F3 (Spearman's $r = -0.53$, FDR adjusted $p = 0.03$), F4 (Spearman's $r = -0.57$, FDR adjusted $p = 0.025$), F7 (Spearman's $r = -0.55$, FDR adjusted $p = 0.025$), and F8 (Spearman's $r = -0.62$, FDR adjusted $p = 0.025$).

Significant associations between EEG–BNST coherence and HAMD improvement were observed exclusively for the right BNST at multiple EEG sites (Fz: Spearman's $r = -0.58$, FDR adjusted $p = 0.043$; F3: Spearman's $r = -0.66$, FDR adjusted $p = 0.043$; F4: Spearman's $r = -0.60$, FDR adjusted $p = 0.043$; F7: Spearman's $r = -0.60$, FDR adjusted $p = 0.043$; F8: Spearman's $r = -0.59$, FDR adjusted $p = 0.043$). Although coherence with the right BNST also correlated with MADRS and HAMA improvements, these did not survive correction for multiple comparisons. No significant associations were observed for the left BNST (Supplementary Fig. 5c).

## The utility of BNST theta biomarker in longitudinal dynamic tracking

Due to the predictive nature of BNST theta-band power for BNST-NAc DBS efficacy, we speculated that DBS may show therapeutic effect by modulating theta-band activity. To elucidate this, we wirelessly streamed multiple 5-min episodes of intracranial BNST electrophysiological data from 10 patients over the randomized crossover phase.

We aimed to determine if high-frequency active versus sham DBS could modulate BNST theta-band activity in keeping with their effects on depressive symptoms. Hence, we aligned daily LFP recordings with corresponding HAMD scores and calculated the mean BNST theta-band power for each day. During the washout period, we averaged BNST theta-band power and HAMD scores to account for potential prolonged effects of BNST-NAc DBS. VAS scores and BNST theta power were normalized to enable comparison between patients. We showed that BNST theta-band activity correlated significantly with HAMD scores (Spearman's $r = 0.42$, $p = 0.027$) (Fig. 4a). Furthermore, active BNST-NAc DBS significantly reduced BNST theta-band activity from the baseline perioperative off stimulation state compared to sham stimulation (Wilcoxon signed-rank sum test, $z = -2.55$, FDR adjusted $p = 0.04$) (Fig. 4b).

To further demonstrate the utility and feasibility of BNST theta power to track subjective emotional dynamics beyond measures, we

labeled LFP episodes with momentary assessment of mood and anxiety states. Across 9 patients (110 resting state datasets), BNST theta power correlated positively with anxiety ratings (Spearman's $r = 0.24$, $p = 0.012$) but not with mood ratings ($p > 0.05$), while anxiety and mood ratings positively correlated with each other (Spearman's $r = 0.369$, $p < 0.001$) (Fig. 4c).

## Neuropsychological predictive biomarker: negative emotional bias

Patients with MDD commonly exhibit biased processing of emotional stimuli, linked to dysfunction in the prefrontal-BNST network[31,32]. We hypothesized that the predictive theta biomarker may underlie this negative emotional bias in MDD. To investigate, we assessed subjective emotional bias, identifying subjective emotional valence and arousal in response to visual emotional stimuli at the perioperative baseline (the physiological results of this task-based measure are reported separately).

We reaffirmed previous findings indicating that higher neutral-controlled negative arousal correlated with more severe baseline depression/anxiety (HAMD: Spearman's $r = 0.69$, $p = 0.002$; MADRS: Spearman's $r = 0.56$, $p = 0.021$; HAMA: Spearman's $r = 0.62$, $p = 0.008$; all survived after FDR correction), with no significant correlation with baseline anhedonia ($p > 0.05$) (Fig. 3c)[31,33].

On an exploratory basis, we found a state-related baseline behavioral measure, negative emotional arousal bias ((positive-negative)/neutral)), which was correlated with resting-state BNST theta power (Spearman's $r = -0.52$, unadjusted $p = 0.037$) and coherence (Spearman's $r = -0.53$, unadjusted $p = 0.029$) (Fig. 3c). In other words, lower negative emotional bias or lower arousal to negative versus positive images correlated with lower theta power. Furthermore, this lower bias also predicted better 3-month improvements of all three core symptoms (DARS: Spearman's $r = -0.48$, $p = 0.052$; MADRS: Spearman's $r = 0.56$, $p = 0.02$; HAMA: Spearman's $r = 0.51$, $p = 0.036$; all unadjusted), suggesting its comparable predictive utility (Fig. 3c). In contrast, baseline clinical questionnaires did not correlate with baseline physiology or predict treatment outcomes.

We analyzed if BNST theta power or coherence mediated the capacity for negative emotional bias to predict clinical improvement using a mediation model. BNST theta power mediated the relationship between 3-month anxiety improvement and negative emotional bias. The direct effect of negative emotional bias on anxiety improvement was insignificant. However, there was a significant indirect effect ($b = 0.31$, 95%CI: [0.085, 1.198]) via BNST theta power (negative emotional bias to BNST theta power: $b = -0.49$, $p = 0.02$; BNST theta power to anxiety improvement: $b = -0.63$, $p = 0.003$) (Supplementary Fig. 3b).

## Quality-of-life and disability: contributors and predictors

The cohort observed improved quality-of-life and reduced psychosocial disability after BNST-NAc DBS, along with reductions in depression, anxiety, and anhedonia. We evaluated whether these symptom improvements (HAMD, MADRS, HAMA, DARS) contributed to overall gains in quality-of-life and psychosocial function, applying FDR correction to control for multiple comparisons across these measurements. WHOQOL-BREF improvements were associated with greater anhedonia improvements at 6 months (DARS: Spearman's $r = 0.54$, $p = 0.005$, survived after FDR correction) and with greater improvements in depression, anxiety, and anhedonia at 1 year (HAMD: Spearman's $r = -0.47$, $p = 0.034$; MADRS: Spearman's $r = -0.48$, $p = 0.029$; HAMA: Spearman's $r = -0.52$, $p = 0.016$; DARS: Spearman's $r = 0.46$, $p = 0.036$; all survived after FDR correction). SF-6 improvements were linked to greater anxiety and anhedonia improvements at 3 (HAMA: Spearman's $r = -0.44$, $p = 0.025$; DARS: Spearman's $r = 0.50$, $p = 0.009$; all survived after FDR correction) and anhedonia improvements at 6 months (DARS: Spearman's $r = 0.57$, $p = 0.003$; survived after FDR

correction). SDS improvements correlated with greater anxiety and depression improvements at 3 (HAMA: Spearman's $r = 0.62$, $p < 0.001$; HAMD: Spearman's $r = 0.48$, $p = 0.013$; MADRS: Spearman's $r = 0.64$, $p < 0.001$; all survived after FDR correction) and 6 (HAMA: Spearman's $r = 0.52$, $p = 0.007$; HAMD: Spearman's $r = 0.53$, $p = 0.005$; MADRS: Spearman's $r = 0.64$, $p < 0.001$; all survived after FDR correction) months and improvements in depression, anxiety, and anhedonia at 1 year (HAMA: Spearman's $r = 0.71$, $p < 0.001$; HAMD: Spearman's $r = 0.57$, $p = 0.007$; MADRS: Spearman's $r = 0.67$, $p < 0.001$; DARS: Spearman's $r = -0.44$, $p = 0.047$; all survived after FDR correction) (Supplementary Fig. 4a).

We then investigated whether physiological theta or a neuropsychological biomarker could predict improvements in quality-of-life or disability. Using stepwise regression analysis with predictor selection, we found that the neuropsychological biomarker, negative emotional bias, rather than theta biomarkers, predicted disability outcomes measured by SDS improvements at 3 months ($R^2 = 0.45$, $p = 0.003$), 6 months ($R^2 = 0.37$, $p = 0.009$), and 1 year ($R^2 = 0.36$, $p = 0.017$) (Supplementary Fig. 4b). BNST theta power was associated only with 3-month SDS improvements (Spearman's $r = -0.54$, $p = 0.025$). In contrast, baseline BNST theta power correlated with quality-of-life improvements (SF-36: Spearman's $r = 0.58$, $p = 0.034$; WHOQOL-BREF: Spearman's $r = 0.65$, $p = 0.01$) at 1 year, but not at 3 or 6 months.

## Discussion

To summarize, we identified state independent, objective theta band electrophysiological biomarkers within the prefrontal-BNST network predictive of depression therapeutic outcomes. Lower BNST and prefrontal theta and prefrontal-BNST coherence predicted better longitudinal therapeutic outcomes for depression and anxiety but not anhedonia, and were right-lateralized. These findings were validated across eyes-open and eyes-closed states and corroborated in a longitudinal data set tracking within-subject emotion dynamics. Chronic BNST-NAc DBS further decreased BNST theta activity. Physiologically guided structural connectivity implicated a prefrontal-BNST network focusing on BNST theta power and prefrontal-BNST coherence. These biomarkers, linking a psychological measure of negative arousal bias and anxiety, offer predictive insights into patients' overall well-being following DBS. Thus, our findings highlight an objective prefrontal-BNST physiological biomarker and psychological markers for predicting depression outcomes, emphasizing phenotype differentiation and sensitivity to specific circuits and anxiety-associated biotype.

To date, advances in MDD DBS studies have introduced behavioral and biometric (e.g., facial expression, speech, heart rate, or heart rate variability), neuroimaging (e.g., tractography), and neurophysiological markers (e.g., SCC beta power, aperiodic 1/f activity) to inform symptom severity, treatment response, and remission[4,34–38]. Multiple critical advances have informed biomarker identification in MDD DBS. Recent studies emphasize a key role for precision targeting and individual tractography of the convergence of crucial pathways within the SCC and the integrity of the cingulum bundle[39,40]. Critical studies highlight physiological predictive biomarkers to separate out the disease illness state versus the well state, implicating lateralized beta and gamma power with relevance for tracking relapse[34]. Personalized approaches through multisource stereotactic EEG recordings highlight a role for recording amygdala reactivity and stimulation through ventral capsule connecting fibers[41,42]. Intracranial recordings have also identified alpha desynchronization related to emotional imagery in the subthalamic nucleus and habenula, which shift emotional valence bias with time-locked alpha-specific stimulation[43,44]. While these biomarkers hold strong translational potential, the present study advances the field in three key areas: First, it integrates multimodal approaches to improve the robustness and clarity of biomarker identification. Second, it identifies circuit-specific biomarkers in the BNST

and prefrontal cortex linked to anxiety biotypes, aiding in depression subtype differentiation, personalized treatment decision, and targeted symptom monitoring. Third, it establishes these biomarkers through cutting-edge longitudinal LFP recordings as tracking transient mood states in MDD patients, further validating their role in guiding DBS therapeutic responses and potential as a biomarker for closed-loop stimulation.

This study shows an objective biomarker highlighting BNST theta and prefrontal-BNST coherence predictive of therapeutic outcome following BNST-NAc chronic stimulation. The BNST acts as an integrated station to orchestrate divergent motivational, arousal, and emotional states[16,17,20]. Human electrophysiology and neuromodulation studies support a prominent role of theta oscillation in the PFC-limbic circuit underlying depression physiopathology[45]. Rodent studies suggest that theta and gamma oscillations and theta synchrony in a negative affective circuit underlie relevant processes of aversive anticipation, learned fear, and social avoidance[46].

We further use physiology-guided tractography highlighting a role for ventral internal capsule tracts traversing within the stimulated area around the BNST, implicating the dorsal anterior cingulate and lateral inferior frontal cortex. The terminus points of the identified tracts converge with optimal therapeutic outcomes identified previously from our MDD sample and from convergent DBS studies for OCD targeting the ventral internal capsule, subthalamic nucleus, and BNST[47,48]. While OCD and MDD involve overlapping symptomatology and cognitive-affective networks, the convergence of tracts across disorders and multiple surgical targets underscores the importance of symptom-specific connectivity profiles in optimizing DBS outcomes. Although we have not statistically compared our identified projections, our identified tracts appear to have a more dorsal posterior projection in the anterior cingulate than tracts identified as the optimal therapeutic outcomes for OCD DBS targeting the ventral internal capsule[48]. This aligns with the dorsal-ventral gradient of cortico-striatal circuits, where dorsal pathways regulate affective processes in MDD and ventral pathways mediate response inhibition and compulsive habit formation in OCD[49,50].

We highlight baseline negative emotional arousal bias as a neuropsychological biomarker predicting BNST-NAc DBS response, mediated by BNST theta activity. An emotional information processing bias has been implicated in neural mechanisms underlying depression, involving networks of emotional salience and attention (e.g., amygdala, insula) and regions responsible for emotion monitoring, evaluation, and regulation (e.g., anterior cingulate, dorsolateral prefrontal cortex)[31]. This bias has been shown to be a behavioral predictor of treatment efficacy in mood disorders, with a positive bias anticipating better outcomes[51]. Reversing this bias has been suggested to be central to the antidepressant effects of pharmacotherapy[52–54]. A preference for positive information is linked to improved emotional regulation and rephrasing capacity, thus perhaps enhancing the capacity for positive behavioral reinforcement and augmenting psychotherapeutic approaches following DBS treatment[55]. This preference may also reflect greater neuroplasticity in depression pathology[56]. This neuropsychological approach offers a clinically practical, sensitive method for assessing depressive states and predicting outcomes, particularly in refractory depression requiring DBS, where conventional scales lack discriminative power due to ceiling effects. Together, these findings highlight a role for a prefrontal-BNST physiological network and a neuropsychological predictor of antidepressant outcomes from chronic DBS of the BNST-NAc.

Interestingly, the BNST theta biomarker and psychological marker may be anxiety associated biotype-specific. Evidence for the role of BNST in anxiety is robust, with BNST activity underlying unpredictable threat, social stress, contextual fear, and startle responses characteristic of anxiety and fear disorders[57]. BNST DBS has also shown efficacy in treating OCD and depression with high comorbidity with anxiety

symptoms[24,26,58]. Previous research suggest ventral internal capsule DBS may be effective for depression with anhedonia, while subcallosal cingulate DBS may better address depression characterized by sadness or mental anguish[59]. These findings underscore the importance of distinguishing depression biotypes addressing the heterogeneity of depression and personalized DBS targets, circuits, and stimulation parameters to individual phenotypes or biomarkers.

Additionally, BNST theta activity correlated with improvements in HAMD, MADRS, and HAMA scores but specifically tracked momentary anxiety changes. These differing measures represent the different temporal dimensions of the mood and anxiety. Clinical rating scales (e.g., HAMD, MADRS, and HAMA) assess symptoms over the past two weeks and reflect general levels of anxiety or mood, often considered pathological traits. In contrast, ecological momentary VAS ratings capture fluctuating emotional dynamics, reflecting transient states[60]. While depression and anxiety severity are often highly correlated in MDD patients, the dissociation between anxiety and mood states in BNST physiomarkers in this study may reflect heterogeneous symptomatology and physiopathology across timescales, with transient states introducing greater variability and context dependence[61]. Notably, although momentary VAS ratings are commonly used, validated, objective, and reproducible measures for state anxiety and mood (e.g., heart rate, heart rate variability, skin resistance) are needed to substantiate the findings.

The lateralization of theta biomarkers is clinically important, potentially reflecting circuit-level abnormalities in depression. The underlying mechanism, though unclear, may involve lateralized emotion regulation. Critically, this points to the potential of neuromodulation targeting the right prefrontal–BNST circuit or unilateral BNST DBS as a treatment for depression. This concept is reinforced by a prior proof-of-concept case where closed-loop stimulation of the right VC/VS, with sensing at the right amygdala, successfully eased depressive symptoms[42].

The study is not without limitations. First, our findings in the BNST appear to be predictive of depressive outcomes associated with anxiety but not anhedonia, suggesting potential differential effects on depression subtypes. Further larger studies are indicated to assess the effects on depressive subtypes.

Second, although patients were blinded to active and sham DBS conditions, they could correctly identify stimulation settings. The potential compound effects of stimulation on neural activity and neuroplasticity, coupled with the absence of a defined washout period, raise the possibility of carryover effects. Notably, BNST-theta power remained low in the sham phase for some patients who started the crossover study with stimulation, suggesting a potential carryover effect and highlighting the need for longer washout periods in future studies to mitigate these effects. Moreover, wireless recordings were conducted off-stimulation, and stimulation artifacts were difficult to adequately control, limiting our ability to accurately capture stimulation effects and potentially explaining the divergent relationship between theta and HAMD observed after active or chronic stimulation.

Third, some results were not robust to correction for multiple comparisons, and observed correlations explained limited variability, highlighting the need for validation in independent, larger datasets.

Fourth, E-field models are based on some assumptions about DBS mechanisms that remain uncertain, serving as approximations rather than definitive representations of its effects on surrounding tissues[62]. While normative connectomes are validated for estimating connectivity, they may not capture individual differences crucial for optimizing outcomes[63–65]. We note that although we used on-stimulation E-field model estimation to link perioperative baseline physiology and fiber tracts, we did not examine the effects of stimulation on physiology, thus limiting our capacity to fully assess physiology-guided tractography.

Fifth, the limited sample size prevented rigorous cross-subject validation, and k-fold cross-validation may have led to data leakage. Therefore, generalizability across participants remains unproven. Future studies with larger cohorts and leave-subject-out validation are needed.

Finally, decoupling of neuropsychological task and neural recordings limits direct insights into their relationship, highlighting the need for task-based neurophysiological studies.

Our study identifies a robust state-independent predictive biomarker within the prefrontal-BNST circuit. These findings offer direct insights into the role of the prefrontal-BNST circuit in human MDD pathology. Our findings suggest a preoperative EEG physiological and psychological negative emotional bias prognostic baseline biomarker with the potential to help guide patient stratification and DBS target selection for subgroups sensitive to BNST-NAc DBS. Subsequent intracranial and EEG physiology might help guide stimulation optimization. Further work integrating multimodal predictors is recommended to better predict both symptom and psychosocial functional outcomes. Given the capacity of BNST theta to track inter-individual mood states, this may be a promising biomarker, aligning with the growing interest in closed-loop therapeutic strategies for neuropsychiatric disorders. Our study also has potential implications for non-invasive stimulation protocols. It sheds light on the antidepressant benefit arising from theta burst stimulation over prefrontal areas through top-down regulation[66]. The identified fiber pathways are relevant to the novel repetitive transcranial magnetic stimulation protocols targeting the orbitofrontal cortex as an alternative or augmentation for treatment-resistant depression[67–69]. Moreover, our findings pave the way for a personalized, circuit-selective, and frequency-based approach for MDD across available multiple intervention modalities. Thus, we highlight identification of predictive biomarkers for treatment with a comprehensive multimodal framework providing a generalizable and extensible approach for biomarker research in translational medicine.

## Methods
### Study participants
Subjects were recruited from Ruijin Hospital, Shanghai Jiaotong University School of Medicine (Shanghai, China). Inclusion criteria for the surgery were: (1) men and women aged 18–65 years; (2) meeting the international Classification of diseases−10 definition of non-psychotic MDD, (3) current episode ≥2 years duration and/or more than 4 repeated episodes with current episode ≥1 year duration and a minimum of 5 years since the onset of the first depressive episode[14], (4) lack of antidepressant response to a minimum of three antidepressant treatments of adequate dose and duration, including at least two medications from two different classes, and failure of adequate psychotherapy or electroconvulsive therapy (either poor response, intolerance or rejection), (5) remaining stable with the current anti-depressive medicine for the last month and (6) Hamilton Depression Scale-17 (HAMD-17) score ≥17. Exclusion criteria were: (1) schizophrenia or psychosis unrelated to MDD; (2) severe personality disorder, neurological disorders, or medical disorders; (3) history of brain surgery; (4) contraindications for anesthesia or stereotactic surgery. We enrolled 26 patients with MDD in the study. All included patients provided written informed consent for participation and for publication of potentially identifiable information. Gender and ethics were self-reported by patients.

The study was conducted in Ruijin Hospital, Shanghai Jiaotong University School of Medicine (ClinicalTrials.gov identifier: NCT04530942). Approval for the trial was obtained from the Ruijin Hospital Ethics Committee, Shanghai JiaoTong University School of Medicine, under the approval number 2021-52. The research use of the implanted components authorized by the Shanghai Testing & Inspection Institute for Medical Devices and approved by China Food and

Drug Administration. A total of twenty-six patients were enrolled in the study from March 29th, 2021, to July 28th, 2023.

## Surgical procedure and stimulation

All patients received bilateral implants of 8 contact cylindrical electrodes (lead model SR1202-S; SceneRay, Suzhou, China) using stereotactic frame-based, magnetic resonance localization with the middle contact point in the BNST and a lower contact point in or close to the nucleus accumbens (NAc). The length of each contact is 1.5 mm and the spacing between contacts is 0.5 mm. Target coordinates for the electrode tip were approximately 4–8 mm lateral to the midline, 1–3 mm anterior to the anterior border of the anterior commissure and 5–8 mm inferior to the anterior commissure. Electrodes were connected via subcutaneous extensions to a stimulator (SR1103, SceneRay).

DBS settings were initiated one-week post-lead implantation, with parameter selection guided by immediate responses to stimulation and careful monitoring for transient side effects. Stimulation parameters, including electrode contacts, voltage, pulse width, and frequency, were systematically adjusted approximately every two weeks. Monopolar stimulation configuration was predominantly utilized across the patient cohort. In practice, we increased the amplitude upon observing diminished beneficial effects or limited clinical improvement, potentially reaching 6 mA. Subsequently, we tailored frequency and pulse width as needed. If these adjustments were ineffective, we introduced the second and, if required, the third pre-selected contact for monopolar stimulation.

## Study design of phase II trial

The study started with a longitudinal, open-label trial followed by a randomized, double-blind crossover design. Following electrode implantation, patients underwent an initial open phase lasting at least six months, during which they were assessed monthly for symptom severity, with adjustments made to their parameters approximately every two weeks. Following the open label phase, patients were randomly assigned to either receive active DBS for two weeks followed by sham DBS for two weeks (the on-off group), or vice versa (the off-on group). Prior to each active and sham phase, a 2-day washout period was implemented. If deemed necessary by the research team or at the request of the patients, the phase was terminated, with patients progressing to the next phase upon termination. Throughout the crossover phase, medication and DBS parameters remained constant. Patients underwent evaluation at five time points: pre-randomization, following two washout phases, and after each active or sham DBS phase.

## Clinical assessments

Depression symptoms were assessed using the HAMD-17, with scores ranging from 0 to 52, where higher scores indicated more severe symptoms[70]. A patient was considered a responder if they exhibited a decrease of at least 50% in their HAMD score compared to baseline. Remission was defined as achieving a HAMD-17 score of less than 8. The HAMD-17, developed for patients with major depression assesses depressive, anxiou,s and somatic symptoms and is considered multidimensional[71]. MADRS, a unifactorial measure assessing primary sadness or lassitude and more sensitive to treatment changes, was used for complementary depression evaluation[72]. The MADRS ranges from 0 to 60, with higher scores indicating more severe symptoms[73]. Anxiety severity was evaluated using the Hamilton Anxiety Scale (HAMA), which has a range of 0–56[74]. Anhedonia was self-reported using the DARS, ranging from 17 to 85, with higher scores indicating lower levels of anhedonia. Additional outcomes included quality of life, assessed via World Health Organization Quality of Life-BREF (WHO-QOL-BREF) and 36-Item Short Form Survey Instrument (SF-36), and functional disability, evaluated using the SDS.

## Affective task

Emotional behavior exhibits a structured organization across two psychophysiological dimensions: valence, varying from negative to positive, and arousal, varying from low to high. The individual assessment of these dimensions can be achieved via an affective task[75], wherein patients were asked to self-report valence and arousal in response to affective imagery. The affective imagery was sourced from the well-established International Affective Picture System and presented in three distinct conditions: positive, neutral, and negative[76]. Each condition encompassed a set of ten different images, with half rated for valence and half rated for arousal. Following the presentation of each image, participants rated their valence and arousal levels using sliding visual analog scales (VAS) that ranged from 0 to 100. A score of 0 represented very negative (valence) or not exciting at all (arousal), while a score of 100 represented very positive (valence) or very exciting (arousal). The intertrial interval lasted between 1 and 1.5 s. The affective task was performed at baseline perioperative visit.

## Localization

DBS electrode localization for the purposes of physiological analyses utilized Lead-DBS v2 (http://www.lead-dbs.org)[29]. In brief, postoperative CT images were first linearly co-registered to preoperative MRI and normalized into ICBM 2009b NLIN asymmetric space using the SyN approach implemented in Advanced Normalization Tools. DBS electrodes were then localized using Lead-DBS and warped into the Montreal Neurological Institute (MNI) space using the PaCER algorithm after visual review and refinement of the co-registrations and normalizations.

## Volume of tissue activated and connectivity estimation

On an exploratory basis, we sought to link this baseline perioperative physiology of BNST theta and EEG-BNST theta coherence, which was clinically predictive of therapeutic outcome with fiber tracts activated at 3-month stimulation (following open-label parameter adjustments tailored to patient clinical outcomes, mean [SD] amplitude: 4.45 [0.4] V) that were associated with perioperative baseline BNST theta and EEG-BNST theta coherence. We speculated that these associated fiber tracts may represent connectivity pathways relevant to the identified physiology and therapeutic effect of DBS, with perioperative baseline BNST theta or EEG-BNST theta coherence acting as a potential predictor of target engagement.

We applied the DBS fiber filtering approach as introduced by Baldermann et al.'s[77]. Specifically, the E-Field was estimated using a finite element method on a four-compartment mesh describing local grey and white matter, as well as electrode contact and insulating material. Subsequently, voxel-wised normative structural connectivity seeding from bilateral E-fields were estimated using normative data sets from the Human Connectome Project at Massachusetts General Hospital (32 subjects, multi-shell diffusion-weighted imaging data).

E-field values were used as weights to construct structural connectivity profiles. For each patient, fibers passing through a non-zero voxel of the E-field were extracted from the normative connectome and mapped onto a standardized voxelized volume with 2 mm resolution. Each fiber received the weight of the maximal E-field magnitude of its passage and fiber densities were weighted by these values.

Each fiber was assigned an R-value based on the Spearman correlation between its weighting and baseline BNST theta or EEG-BNST theta coherence. A high R-value indicated that the baseline physiology is associated with the engagement of the tract. Significant (unadjusted) tracts were then integrated to refine the physiological correlates.

## Perioperative signal recordings

Twenty-six patients underwent perioperative signal recording. Nine patients were excluded: three due to data noise exceeding 50%, three for incomplete 3-month follow-up during analysis, and three because

contacts did not meet BNST targeting criteria for this stringent physiological analysis. Consequently, the reported results are based on the remaining 17 patients.

The surgery followed a staged procedure. We recorded signals between 2 and 4 days postoperatively, during the period when the leads were externalized. Resting state recordings were obtained for a minimum of 3 min (maximum 5 min) while patients sat in a relaxed and wakeful state. To assess potential sensory-attentional state effects on brain activity, recordings were conducted under two conditions: eyes-open and eyes closed.

Scalp electroencephalogram (EEG) and local field potential (LFP) data were simultaneously recorded using a BrainAmp MR amplifier (Brain Products) at a 500 Hz sample rate. EEG was obtained from 7 frontal electrodes (Fp1, Fp2, F3, F4, F7, F8, Fz) using the 10–20 placement system. We focused on frontal electrodes due to their critical role in affective processing and relevance to prefrontal-BNST connectivity, essential for mood and anxiety regulation[78]. Moreover, the surgical bandages over the forehead limited electrode placement options, and restricting the number of electrodes helped minimize patient discomfort during postoperative recordings. To ensure that the desired signals were not contaminated by blinks or saccades, electrooculogram (EOG) recordings were performed. The data was reference online using a left mastoid electrode. The ground was placed at the apex nasi.

## Wireless data streaming

Ten patients participated in the daily in-lab electrophysiological recordings and momentary self-report behavioral assessments during the randomized crossover phase. One patient was excluded due to insufficient data available during the active stimulation phase, leaving a total of nine patients included in the final analysis.

Patients completed 1–3 trials per day during the pre- and post-active and sham stimulation phases, while stimulation was turned off. In each trial, participants engaged in 5 min of wireless data streaming, followed by self-reporting on their mood or anxiety states using a Visual Analog Scale (VAS). Patients maintained an open-eye resting state throughout the trial. The intertrial interval was more than 4 h.

Wireless data streaming was performed via the IPG while in-clinic with a sample rate of 415 Hz or 1000 Hz (for P3). LFPs were sensed in bipolar configuration, where a pair of sensing contacts were recorded with one contact referenced to the other. The contact pairs within (defined as less than 1 mm of distance from the target) or closest to the BNST were preselected for recording. Patients rated their mood or anxiety states on the VAS immediately after data streaming, where 0 corresponds to "very happy" or "not anxious at all", 100 corresponds to "very sad" or "very anxious", and 50 corresponds to "moderate state".

## Signal processing

All data were analyzed offline using MATLAB R2023a with EEGLAB toolbox v2022.1 and custom-written scripts.

Perioperative LFP data applied offline bipolar referencing to reduce volume conduction effects. The data were high-pass filtered at 1 Hz using a 2nd order Butterworth filter (*pop_basicfilter*) and band-stop filtered at 50 Hz and its harmonics (*pop_eegfiltnew*). Independent Component Analysis, including EOG for identifying blinks/saccades, followed by visual inspection, were utilized for artifact rejection. Data epochs or electrodes containing artefactual activity (e.g., blinks, saccades, or cardiac activity or line noises) identified during visual inspection were omitted from further analyses. For eyes-closed datasets, $62 \pm 50$ s (range: 4–208 s) were removed; for eyes-open datasets, $48 \pm 40$ s (range: 0–141 s) were excluded. The final analysis included 17 patients' eyes-closed datasets ($230 \pm 55$ s, range: 140–308 s) and 15 patients' eyes-open datasets ($227 \pm 52$ s, range: 94–282 s).

Based on the atlas-defined lead reconstruction in the MNI space using Lead-DBS v2 toolbox, the LFP contact pairs within or closest to the one of the three brain structures (BNST, NAc, hypothalamus) was selected from each recorded hemisphere for final analysis. The BNST and hypothalamus contacts ($n = 17$) were all within or <1 mm from the closest voxel within the structures. The NAc contacts ($n = 11$) were within <2 mm distance. To facilitate subsequent individual-level analysis, we computed the LFP activity in the BNST, NAc, and hypothalamus by averaging across both hemispheres. EEG data were averaged across 7 channels.

Wireless LFPs followed a preprocessing procedure akin to perioperative LFPs, with the addition of band-stop filtering at 40 Hz and its harmonics (*pop_eegfiltnew*) to cancel out the sensing noise. To maintain consistency, wireless data obtained from P3 were down-sampled at 415 Hz.

Spectral analysis was performed using the Welch's power spectrum density estimation (*pwelch*). Non-overlapping windows of 512 data points were analyzed, affording a spectral resolution at 0.977 Hz. LFP power at frequencies <4 Hz was excluded from further analysis to eliminate the movement artifacts and the influence of 1/f nature of the signal. Our primary area of interest was power within the 4–90 Hz range. To mitigate the effects of individual variance in targeting and impedance, we computed the relative (%) powers by expressing the power within this range as a percentage of the total power. We summed power into canonical frequency bands: θ (4–8 Hz), α (8–12 Hz), lβ (13–20 Hz), hβ (21–35 Hz), γ (40–90 Hz).

We fit spectral power features of regions or functional connectivity between regions (spectral power: 2 regions × 5 bands; coherence: 5 bands; total 15 features) into a penalized ridge regression model with permutation feature selection to identify the electrophysiological features predictive of clinical improvements. The modelling pipeline consisted of built-in threefold (without overlap) cross-validation for model hyperparameter optimization and repeated 10000 iterations for stability and better performance. Permutation feature importance was then computed for the model based on the decrease in a model score when a single feature value is randomly shuffled. The model was trained on 3-month improvement data and applied to 6- and 12-month outcomes, with $R^2$ scores reported for model performance. Due to the small sample size, these $R^2$ scores were not cross-validated.

Non-directional spectral connectivity was computed using coherence analysis. The continuous data were divided into 10-s epochs with a 50% stepwise overlap. The magnitude-squared coherence was then calculated for each epoch using the Welch's overlapped averaged periodogram method with blocks of 512 samples, a Hanning window, a 50% overlap, and frequency resolution of 0.977 Hz in 4–90 Hz range (*mscohere*) and scaled by 100 for expressing percentages. The coherence values were summed across frequency bins and then averaged across epochs. These averaged values were subsequently grouped into five predefined frequency bands.

Directional spectral connectivity (estimates of both A - > B and B - > A connectivity) was estimated using spectrally resolved Granger causality[79]. This analysis utilized the MVGC multivariate Granger causality toolbox. To maintain data stationarity, LFP signals were initially segmented into 4-s time-windows with 50% overlap. A multivariate vector autoregressive (MVAR) model was fit to the full "universe of data" time series, with model order optimized using Bayesian information criterion. The corresponding MVAR model parameters for the selected model order was estimated and the resulting model was checked for stability. Then, the frequency conditional spectral causalities were computed for signal pairs and the significance of the resulting causalities was tested via nonparametric permutation testing. The subsequent analysis focused on the summed frequency-based connectivity within the theta (4–8 Hz) band, based on the prior findings.

Granger causality can be sensitive to differences in signal-to-noise ratios between time series, potentially leading to spurious directionality. To address this, we computed Granger causality for time-reversed signals and compared it to the non-reversed standard results[80]. Time reversal was expected to reverse the direction of true causality while preserving spurious effects due to confounding between-area power differences or signal-to-noise.

## Statistical analysis

For analysis of the clinical data, two patients (P8, P12) missed assessments at months 3 and three patients (P16, P18, P22) at months 12; we used data from the previous visit the month prior in the analysis. Data are presented as mean (standard deviation, SD) in tables and as mean (standard error, SE) in figures, and statistical significance was set at a 2-tailed 5% level. Data normality was determined by Shapiro−Wilk test. Correlational analyses were performed utilized Spearman rank correlation or Pearson correlation. Group differences were inferred using Wilcoxon signed-rank sum test or paired student $t$-tests. Linear regression was used to model the relationship between a dependent variable and one or more independent variables using the stepwise method. We applied false discovery rate (FDR) correction to account for multiple comparisons where necessary.

Mediation model was used to test the indirect effect of negative emotional bias on clinical outcomes via theta biomarkers[81]. It was performed using Model 4 in the PROCESS Marco (SPSS). Note that the negative emotional bias did not follow a normal distribution and therefore, a pre-applied rank-based inverse normal transformation was performed. The mediation model investigated two paths: the indirect path from (a) negative emotional bias to theta biomarkers and (b) theta biomarkers to clinical outcomes, as well as the direct path from (c) negative emotional bias to clinical outcomes. Model fit was evaluated based on a non-significant $X^2$ ($p > 0.05$). The indirect effects were estimated using 5000-sample bootstrap procedure with bias-adjusted confidence intervals.

Data analyses were performed using SPSS 26.0, Python 3.11, or MATLAB R2023a. The figures were generated using R version 4.3.1.

## Reporting summary

Further information on research design is available in the Nature Portfolio Reporting Summary linked to this article.

## Data availability

The raw physiological data generated in this study are protected and cannot be deposited in a public repository due to data protection regulations and clinical privacy laws. The data are available under restricted access because they contain sensitive clinical information subject to ethical and legal safeguards. Access can be obtained by submitting a request to the corresponding authors (V.V. or B.S.; E-mail: vv247@cam.ac.uk, Tel: +86 13918129540) together with a brief research proposal. Access is contingent upon approval by the Institutional Review Board (IRB) of Ruijin Hospital, and is limited to qualified researchers at recognized academic or medical institutions for non-commercial scientific purposes. Requests will normally be acknowledged within 2 weeks, with access granted within approximately 6−8 weeks, depending on the IRB review timeline. Approved researchers will be granted access for the duration of their project, renewable if necessary, and must sign a Data Use Agreement specifying restrictions on redistribution, reuse, and authorship requirements. Source data are provided with this paper.

## Code availability

The code used to produce the results in this paper is available at https://github.com/Fallenworm/BNSTDBS[82].

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

## Acknowledgements

This study is funded by the STI 2030-Major Projects (No. 2021ZD0200407 [to V.V.]), the National Natural Science Foundation of China (Grant No. 82271515 [to B.S.], 81971294 [to D.L.], 8240051532 [to Y.W.] and T2250710686 [to V.V.]), the Science and Technology Commission of Shanghai Municipality (Grant No. 20410712000 [to D.L.]), Medical Research Council Project Grant (Grant No. MR/W020408/1 [to V.V.]), the SJTU Trans-med Awards Research (Grant No. 2019015 [to B.S.]), the Scientific and technological innovation action plan of Shanghai (Grant No. KY20211478 [to B.S.]), the Shanghai Municipal Science and Technology Major Project (Grant No. 2021SHZDZX [to B.S.]), and the Nursing Development Program of Shanghai Jiao Tong University School of Medicine (Grant No. SJTUHLXK2022 [to X.Q.]).

## Author contributions

B.S. and V.V. initiated this work. B.S. and V.V. supervised the study. L.W., Y.Y.Z., and V.V. drafted the manuscript. L.W., Y.Y.Z., Q.D., Y.J.Z., Y.W., L.D., K.Y., J.H., X.Z., P.H., X.Q., D.L., S.Z., W.L., and Y.P. collected data for the study. V.V., L.W., B.S., Y.L., and X.L. designed the study. L.W. and Y.Y.Z. conducted the statistical analysis and verified the data reported in the manuscript. L.W. and Q.D. performed the imaging analysis. L.W., S.S., A.M., and V.V. designed the behavioral task. L.W. and L.M. set up the perioperative and wireless signal recording platform. All authors contributed to finalizing the manuscript and reviewed the final version. L.W., Y.Y.Z., and Y.W. are equal contributors listed as first coauthors. B.S. and V.V. are corresponding authors.

## Competing interests

The authors declare no competing interests.
