## [Transparent Peer Review file · Nature Communications]

Prefrontal – Bed Nucleus of the Stria Terminalis Physiological and Neuropsychological Biomarkers Predict Therapeutic Outcomes in Depression

Corresponding Author: Professor Valerie Voon

Version 0:

Reviewer comments:

Reviewer #1

(Remarks to the Author)

Overview

Thank you for the opportunity to review this very interesting paper studying neurophysiological and neuropsychological biomarkers of the response to BNST DBS for treatment-resistant depression. Alongside a randomized clinical trial, the authors collected intracranial neural recordings and EEG to study neural activity associated with DBS-induced changes in depression and anxiety, incorporated neuroimaging to try to identify the fiber pathways involved, and assessed affective processing in a neuropsychological task. Overall, the study is well-executed and makes strides toward addressing the critical issue of predicting depression outcomes and moving toward personalized DBS through several novel components (especially the authors' dedication to multimodal techniques). However, I have some reservations regarding the robustness of the statistical results and choice of analyses, perhaps due to often vague descriptions of both methods and results. Additionally, the introduction/discussion are relatively limited in their scope of the field and literature review. More detail on these comments are provided below, as well as minor edits/suggestions for improving the manuscript.

General Comments

- The introduction is rather limited in its scope. It would be helpful for readers not in the field to add some more context about DBS for depression, including a brief summary of its background/history and how their BNST target fits into the story.

- Overall this study seems like a blend of results of the clinical trial (which the authors state has been submitted for publication elsewhere) and the biomarker discovery analysis (which is the main focus of the study). Some of the results presented, such as the outcomes, the many correlations across baseline, 3mo, 6mo, 12mo scales (in Fig. 2), and the QoL analyses seem better suited for the main clinical trial results, unless the authors can more directly tie these clinical outcome analyses to the present biomarker study.

- Although the authors note several times that their results are cross-validated, these methods are not entirely explained. The two instances of "cross-validation" (in my understanding) are:

(1) eyes-open versus eyes-closed -- this comparison studies state-specific effects, which is important and I commend the authors for thinking about this, but not really cross-validation because the data are still from the same groups of patients (except 2 patients excluded from eyes-closed). I suggest this should instead be framed as a method to test for the robustness of the biomarker under different state conditions, and the results should be clearly stated what was similar vs different across states.

(2) cross-validation in ridge regression – the authors state they performed 3-fold cross-validation here, and this is great but the methods do not explain details of this process (i.e., Were they 3 different groups or was there overlap in iterations? How was performance assessed and reported? Was the model trained on 3mo data?). Although their results appear to hold up to this cross-validation, I caution the authors to claim their biomarker is generalizable given that it is tested here in a limited single-center RCT cohort and should be validated in independent dataset(s) to truly test its generalizability.

- Along these lines with the ridge regression, the rationale for the statistical approach presented in Fig. 3 is not clear. It

appears the authors (a) performed univariate correlations to test relationships between spectral power in various frequency bands and outcomes (is this % improvement?), then (b) performed cross-validated ridge regression (reporting good R2 values but are these the cross-validated values?), and then (c) performed linear regressions of only theta power and coherence over all time points. Why not use the ridge regression model from the 3mo outcomes and try to apply it to 6mo and 12mo?

- Which then brings me to a critical question about correction for multiple comparisons, especially given the vast number of comparisons and correlations performed in this study. In the methods, the authors state they used FDR where appropriate. It appears to me that many of the p-values are uncorrected, but often key information is missing. Throughout the results, the test statistics and p-values should be reported (e.g., no test statistics in lines 108-111) and it should be very clear 1) whether correction for multiple comparisons was performed and 2) if the reported results survived correction for multiple comparisons.

- The physiology-guided tractography, although very interesting and an important anatomical component, is not sufficiently explained and thus difficult to interpret. The VTA model estimates the effect of DBS on surrounding brain tissue and requires stimulation parameters as input. Yet, to my understanding, all neurophysiological recordings were performed off-stimulation. What stimulation parameters were used here? And what neurophysiological variables were assigned to each VTA for the analysis? Further details of these methods, as well as a more accurate description of exactly what the results represent are needed. If there is a disconnect between stimulation and physiology, this needs to be described and discussed as a limitation. Additionally, was modulation of the identified pathways associated with better depression/anxiety improvement in this cohort (briefly mentioned in another cohort I think in lines 283-285 but does it also apply here?). This would greatly strengthen the results. Along these lines, were there no tracts associated with worse outcomes? And did theta power also vary depending on contact location relative to the BNST or other structures?

- Given the tractography results, I am curious if the authors investigated if their EEG results aligned with the anatomical localization of these tracts. If I understand correctly, EEG signals were aggregated across prefrontal channels, but was there any anatomical specificity for specific EEG channels observed for the prefrontal-BNST theta coherence?

- Throughout the methods and results, the authors tend to use somewhat vague terms for variables assessed (e.g., "efficacy for depression" (line 127). While I understand wanting to keep it simple, it is also important to explain exactly what is being assessed (e.g., change in HAMD and MADRS scores at 3 months compared to preoperative baseline). At times it is difficult to tell whether the analysis focused on scores at baseline vs improvement scores at follow-up.

- The discussion lacks depth in literature review and would benefit from some expanded discussion on a few points:
(1) Further discussion comparing the present results to previous studies that have investigated biomarkers in DBS for depression would help situate the study's findings within the broader context of the field.
(2) The authors found a very interesting result that their biomarker was linked to anxiety and depression but not anhedonia. Some additional discussion about how/why these biomarkers may be "subtype-specific" and potential implications would be beneficial.
(3) The authors very briefly allude to the potential clinical utility of their findings. More discussion about how these biomarkers could be incorporated into patient-selection, real-time DBS programming, or even closed-loop systems would enhance the clinical relevance of the study. The noninvasive stimulation angle is interesting given the fiber pathways identified – what would be the proposed target and how would the present results inform future protocols?

- The limitations section in the discussion is missing several key limitations to be mentioned:
(1) some results were not robust to correction for multiple comparisons, and correlations only explain a fraction of the variability in outcomes and thus should be further explored in independent datasets
(2) potential compound effects of stimulation on neural activity and neuroplasticity – we may not know the ideal washout period and thus cannot rule out carryover effects across active/sham
(3) the VTA model requires many assumptions about the mechanisms of DBS that are unknown, and thus serves as an estimate of the effect of DBS on surrounding tissues and is not ground truth
(4) normative connectome is a valid approach to estimate connectivity (can cite several other papers out there from Horn et al and others using this approach) but this may not capture individual connectivity differences, which may be important
(5) decoupling of neuropsychological task and neural recordings – future studies could combine these together to directly evaluate their relationship
(6) use of momentary assessments of mood/anxiety vs clinical rating scale scores – "state" vs "trait" could be worth discussing

- The extended figures/tables were not included in the files for download and thus I was not able to evaluate these.

Minor Edits/Comments

Title/Abstract/Introduction

- The title could be reworked to be clearer – the "Physiological Intracranial Prefrontal – Bed Nucleus of the Stria Terminalis" does not really make sense.

- The abstract focuses mainly on the study's results and implications. A brief summary of the key methodological aspects would be helpful for readers as well.

- Line 33 – define DBS acronym (instead defined in sentence soon after)

- Line 37 – "medial prefrontal" should be medial prefrontal cortex for mPFC

- Line 42 – another motivation for this study (which could be added before "This variability hinders...") is the lack of objective

methods to evaluate depression-related symptoms given their interoceptive nature

- Line 47 – preliminary efficacy – should specify if evidence comes from open-label, RCTs, etc. More description of previous literature would be helpful for context
- Line 48-50 – “Compelling target rationale...” – this sentence is a bit awkwardly worded. Could modify to something like “The rationale for BNST as a DBS target is largely based on its involvement in features of depression, including sustained fear, stress responses, social behavior, and valence surveillance.”
- Line 51-52 – missing comma between hypothalamic and autonomic

Results

- Line 53 – typo – “relative” should be “relatively”
- Line 74 – why is the cohort so predominantly male?
- Line 87-90 – were postoperative wireless recordings performed in-clinic?
- Line 91 and Line 98 – what is meant by “baseline” – preoperatively? If so, how soon before surgery?
- Line 92-93 – the rationale for using both the HAMD-17 and MADRS as measures for depression should be explained – how do they complement each other and why do they both need to be included in all analyses?
- Line 101-103 – is this during the open-label phase? Please clarify.
- Line 103-104 – was multisite monopolar stimulation part of the study design, or was this the result of the programming process? Please clarify.
- Line 115-117 – “Notably at a cohort level...” – this sentence is confusing. I think it would be clearer to say something like “At preoperative baseline, anxiety and depression were highly related (as shown by high correlations between HAMD, MADRS, and HAMA scores), while anhedonia was not correlated with depression or anxiety scores, indicating potentially different clinical phenotypes.”
- Line 115-116 – I’m also unsure whether correlations between depression/anxiety/anhedonia scores are a valid way of discerning phenotypes of response to DBS. Wouldn’t it be clearer to assess the effect of DBS on these symptoms separately? Would disappearance of correlation between two symptoms really indicate a different response?
- Line 127-128 – is “efficacy for depression” defined as change in these scores compare to baseline?
- Line 132 – “we only found predictive effects and no state related effects” – please clarify, does this mean the features were not different between eyes open vs eyes closed?
- Line 133 – unclear what “5-frequency band average power” means?
- Line 146-149 – specifying that the model was trained on eyes-open vs eyes-closed would be helpful. Also throughout manuscript these should be written as “eyes-open” or “eyes-closed”
- Line 155 – “predicted” – be careful, this is not prediction because not validated. Should use “correlated with” instead
- Line 179 – what does “fluctuations in BNST theta-band activity” mean? Do you mean that BNST theta-band activity correlated with HAMD scores?
- Line 181 – was the amount of reduction in theta power correlated with the amount of improvement in depression symptoms? Also please clarify, was theta power reduced compared to baseline off stim recording?
- Line 187-190 – this analysis and rationale is a bit confusing. This comes across like a strange roundabout way of saying theta power was correlated with both anxiety and mood, even though it was only correlated with anxiety. Some discussion about momentary assessments vs classic MADRS/HAMD may also be warranted.
- Line 203-206 – it needs to be more clear that theta power and coherence were not recorded during the task. If I understand correctly, the task preceded the recordings (baseline pre-DBS?) and thus theta may have changed due to compound effects of stimulation?

Discussion

- Line 247 – tone down the “state independent, ecologically valid, and generalizable” descriptors unless the results can be more clearly stated. What is meant by ecologically valid and how do the data support this?
- Line 252 – not a separate data set – rephrase to say across eyes-open and eyes-closed states
- Line 263-271 – citing oscillatory activity in DBS for PD is not critically relevant to the focus of this paper. Instead it would be more beneficial to expand this paragraph to discuss work that relates to biomarkers of depression and DBS (see main comment above)
- Line 276 – what compelling evidence supports the role of theta?
- Line 287-288 – what was the rationale for comparing to the OCD tracts? And why do the authors think that these tracts are more dorsal – what might be the underlying mechanism? More discussion here to generate hypotheses would be useful.
- Line 290-298 – this paragraph repeats results and could instead expand more on previous studies of emotional arousal bias in depression and how it fits in here, as well as neuropsychological biomarker use in the future.
- Line 305 – again, careful about use of “generalizable”

Methods

- Line 443-444 – should “failure of adequate psychotherapy and ECT” be a separate number in this list?
- Line 460-461 – are the contacts all cylindrical or are there also segmented? Please specify
- Line 509 – missing “and” between (SF-36) and functional disability
- Line 515 – could add citation(s) about task validity
- Line 526 – clarify what is meant by baseline – is this preoperative? This needs to be clarified throughout the manuscript as well
- Line 538 – Baldermann et al approach implemented in Lead DBS? See major comment above about this analysis and add info about stimulation and neurophysiology link and the rationale for this. What “identified neurophysiology” was used in this analysis?
- Line 556 – in the three that were unable to record, was this because their stimulation parameters were not bipolar-friendly? Was the sandwich configuration required?

- Line 563-564 – could provide some more rationale for eyes-open vs eyes-closed as assessing state effects
- Line 567-568 – focused only on frontal electrodes – rationale for excluding other brain regions?
- Line 584 – was wireless data streaming performed via the IPG? And were these recordings done while in-clinic? Please clarify
- Line 591 – typo “moderate state”
- Line 599-601 – how many data epochs were omitted? Could mention use of the EOG recordings as well here for identifying blinks/saccades
- Line 609-611 – are these 1mm/2mm distances to the center of the structures or the border?
- Line 620 – can report the exact spectral resolution
- Line 622 – 1/f is a hot topic in the world of biomarkers, including depression (Veerakumar et al, 2019; PMID: 31314668). Did the authors consider this or aperiodic offset as a potential candidate markers in addition to spectral power?
- Line 631 – further details about cross-validation and hyperparameters would be helpful (see main comment above).
- Additionally, what is meant by “treatment response”? Was this 3mo improvement compared to preop?
- Line 636 – “Non-directional” would keep consistent with “Directional” used next for GC analysis
- Line 641 – report the exact spectral resolution
- Line 659 – how many participants were missing assessments? And was data taken from time points before or after the missing time point? Unclear how this is directly relevant to the analysis vs clinical trial results so some clarification would be helpful
- Line 668 – what is meant by “stepwise method” here? It appears that linear regressions were performed with single variable
- Line 670 – could add relevant citation(s) for mediation models, especially use in similar analyses
- Line 677 – typo delete the ‘

Figures

- In general, the figure captions could be more descriptive to aid in interpretation, especially for these complex correlation matrices.
- The use of “Imp” and “Bs” in figures is not super intuitive. Would be helpful for reader if the authors spelled out variables whenever possible to aid in interpretation
- Fig 1: nice overview. Should figure read “N=15 eyes open” to match text? Typo in electroencephalography in caption.
- Fig 2: the rationale for this large correlation matrix and its relevance to the present study is unclear. The takeaway from this figure is rather unclear. If the authors do keep it, I would recommend pulling key correlations for this main figure to demonstrate the main point and instead having a supplementary table with R/p-values for reader to reference if interested
- Fig 3: in panel a, the axes should be labeled with improvement. Also could clarify in caption, neurophysiological signals are from perioperative off-stim correct?
- Fig. 4: a and b in the figure caption should be switched. Panel c is quite the complex figure. More description in the caption would be helpful to know what is the main takeaway readers should be getting from this figure. Are p-values not corrected?
- Fig. 5: Panel a – I am curious about the differences observed between sham/active vs active/sham groups. It appears all patients in sham/active group show the expected relationship between theta and HAMD, compared to only 2/5 in active/sham. What do the authors hypothesize? Could be an interesting discussion point. Additionally, it could be clearer to display all patients with consistent ordering of stages (i.e., post-open label, post-active, post-sham) to aid in easy comparison. In panel b why is the post-open label (stage 1) not included in the group analysis? Also caption for panel b should say various frequency bands instead of just theta power (as all are presented).

(Remarks on code availability)

The code appears to include basic processing of the LFP and EEG data and the ridge regression. It does not include a README. I did not attempt to install or run the code.

Reviewer #2

(Remarks to the Author)

The manuscript by Wang et al. reports biomarker findings from a randomized crossover clinical trial of BNST-NAc DBS for treatment-resistant depression. BNST-NAc DBS is reported to decrease symptoms as measured using a variety of scales. Using perioperative LFP and EEG recordings, the authors demonstrate that BNST theta power and prefrontal-BNST theta coherence predicted the outcome as measured by different rating scales. Using recordings off the DBS device, they demonstrate that BNST theta tracked depression severity longitudinally over the course of the randomized crossover phase of the trial. Additionally, they demonstrate that BNST theta power is correlated with negative emotional arousal bias.

The study is timely, given the increasing interest in device-based treatments for treatment-resistant depression and the scarcity of neurophysiological measurements from targeted regions during treatment. The crossover trial design is an effective way to disentangle placebo effects, and the results from BNST-NAc-VIC neurophysiology is an advance beyond existing studies. Overall, the findings are consistent, indicating that lower theta activity is associated with symptom improvement, possibly driven by changes in anxiety. However, despite the potential impact, it is difficult to be enthusiastic about the current manuscript given the significant number of important details and methodological concerns that are currently lacking. These issues, which I will detail below, would need to be addressed before I could assess the potential impact of this manuscript and make any recommendation about publication.

- Given the emphasis on randomized crossover clinical trial, clinical outcomes have not been described in sufficient detail. It is not clear how many of the participants responded to treatment or how many reached remission. In addition, the results

from the crossover phase of the study have not been detailed at all. Therefore, it is not clear if DBS effects were lost during the sham stimulation arm of the study. Ideally, the depression severity measures during the sham and active weeks should be compared and included in the results.

- It is unclear why the authors chose a 2-day washout period for the crossover phase. Since treatment-resistant depression is generally a slowly changing disorder, as compared to Parkinson' disease, a 2-day washout period might not be sufficient. On a related note, the BNST-theta power remained low in the sham phase for 3/5 patients who started the crossover study with stim, suggesting a carryover effect.

- The details of LFP processing are unclear. Did the authors average the spectra/connectivity measures calculated from individual recordings or the timeseries? (Line 611 – 613) Averaging timeseries seems inappropriate, as this process destroys important temporal information. Also, it is not clear why the authors averaged the left and right hemispheres and the 7 EEG channels, as this process removes spatial information that might be relevant.

- Additionally, it is not clear why the authors expect eyes open and eyes closed resting state LFP to be different enough to be considered as separate datasets. While EEG studies often use this distinction for alpha band activity that is clearly modulated with eyes open/eyes closed, it is not clear if the same differences are observed in BNST, NAc, or VIC LFP.

- It is not clear how many samples of the LFP and EEG features per participant were used in the penalized ridge regression model. Similarly, the details of the cross-validation (e.g. leave-n-subject-out, k-fold) have not been reported.

- Given that EEG and LFP may have differences in signal-to-noise ratio, the inference of directionality from Granger causality may be biased. The authors could implement time-reversed GC to confirm the directionality. See for example: Haufe S, Nikulin VV, Müller KR, Nolte G. A critical assessment of connectivity measures for EEG data: a simulation study. Neuroimage. 2013 Jan 1;64:120-33. doi: 10.1016/j.neuroimage.2012.09.036. Epub 2012 Sep 21. PMID: 23006806.

- Given that the authors demonstrate the BNST-theta marker is strongly associated with anxiety (using VAS) rather than mood, it would be useful to see if the prediction of 3 month and 6 month HAM-D and MADRS scores are driven by predictions of changes in anxiety-related symptoms rather than other symptoms like depressed mood, sadness or lassitude

- As a suggestion for clarity, I would recommend streamlining the results section around the paper's main message regarding the neurophysiological findings. Results that don't directly address this, such as correlations between symptom severity scores (Fig 2b), could be moved to the supplement.

(Remarks on code availability)

Reviewer #3

(Remarks to the Author)

(Remarks on code availability)

Version 1:

Reviewer comments:

Reviewer #1

(Remarks to the Author)

The authors have sufficiently addressed all of my comments.

(Remarks on code availability)

The code appears to include basic processing of the LFP and EEG data and the ridge regression. It does not include a README. I did not attempt to install or run the code.

Reviewer #2

(Remarks to the Author)

The authors have addressed many comments satisfactorily. However, these concerns still remain:

- The k-fold cross-validation procedure described will lead to data leakage, i.e., the model is exposed to patient-specific information. The choice of this cross-validation raises questions about generalizability across patients. The authors will need

to demonstrate similar results with leave-n-subject-out cross-validation or remove references to cross-validation and generalizability across the manuscript and include a discussion of this limitation. Also, the comparison of eyes open and eyes closed data suggests that the biomarker is robust to recording conditions/states, which is very useful information. However, the main question about biomarkers being generalizable is whether they can predict across participants. So, using the term generalizable in the context of eyes open and closed is misleading.

- While the authors provide the rationale for averaging LFP timeseries across the hemispheres, the approach still raises concerns for the following reasons. Averaging timeseries across hemispheres destroys temporal dynamics through constructive and destructive interference. It is recommended to average in the spectral domain to preserve temporal dynamics. Furthermore, there may be hemispheric differences in the regions of interest, given that the correlations with HAMA and MADRS are right-dominant. Therefore, the authors are recommended to re-run analyses without averaging across hemispheres.

(Remarks on code availability)

Reviewer #3

(Remarks to the Author)

(Remarks on code availability)

Version 2:

Reviewer comments:

Reviewer #2

(Remarks to the Author)

I appreciate the authors changing the language to make clear that they have not tested the generalizability of their proposed finding (though I would urge the authors to continue to pursue this, as it is concerning that it was not able to be validated through proper cross validation). That being said, I don't have any concerns with recommending this manuscript for publication after the latest revision.

(Remarks on code availability)

Response to Reviewers for

Predicting Therapeutic Depression Outcomes: Intracranial Recordings of the Prefrontal – Bed Nucleus of the Stria Terminal Network

Linbin Wang^{1,2,3†}, Yingying Zhang^{2†}, Yuhan Wang^{1†}, Qiong Ding³, Luling Dai¹, Kejia Hu¹, Kuanghao Ye¹, Xin lv¹, Xiaoxiao Zhang¹, Alekhya Mandali³, Luis Manssuer³, Saurabh Sonkusare³, Yijie Zhao², Peng Huang¹, Xian Qiu¹, Yixin Pan¹, Yijie Lai¹, Dianyou Li¹, Wei Liu¹, Shikun Zhan¹, Bomin Sun^{1‡}, Valerie Voon^{1,2,3‡}

Corresponding author: Valerie Voon, Email: vv247@cam.ac.uk and Bomin Sun, Email: sbm11224@rjh.com.cn

REVIEWER COMMENTS

Reviewer #1 (Remarks to the Author):

Overview

Thank you for the opportunity to review this very interesting paper studying neurophysiological and neuropsychological biomarkers of the response to BNST DBS for treatment-resistant depression. Alongside a randomized clinical trial, the authors collected intracranial neural recordings and EEG to study neural activity associated with DBS-induced changes in depression and anxiety, incorporated neuroimaging to try to identify the fiber pathways involved, and assessed affective processing in a neuropsychological task. Overall, the study is well-executed and makes strides toward addressing the critical issue of predicting depression outcomes and moving toward personalized DBS through several novel components (especially the authors' dedication to multimodal techniques). However, I have some reservations regarding the robustness of the statistical results and choice of analyses, perhaps due to often vague descriptions of both methods and results. Additionally, the introduction/discussion are relatively limited in their scope of the field and literature review. More detail on these comments is provided below, as well as minor edits/suggestions for improving the manuscript.

General Comments

Q1: The introduction is rather limited in its scope. It would be helpful for readers not in the field to add some more context about DBS for depression, including a brief summary of its background/history and how their BNST target fits into the story.

A1: Thank you for your thoughtful, detailed, and thorough feedback. To provide a more comprehensive context for readers less familiar with DBS for depression, we have expanded and refined the introduction as follows:

"DBS targeting the medial prefrontal cortex (mPFC)-limbic network associated with impaired emotion and reward processing in MDD, have demonstrated potential therapeutic efficacy¹⁻⁴. While open-label and longitudinal studies have shown promise since the first report on subcallosal cingulate cortex (SCC) DBS^{5,6}, randomized controlled trials (RCT) for depression have had mixed results with variability related to issues including targeting precision in the context of individual variability in relevant white matter tracks, study design to allow individual optimization, and the heterogeneity of depression²⁻⁴. In addition to the SCC which targets the intersection of three limbic-relevant white matter tracts, two major randomized trials targeting the nucleus accumbens/ventral internal capsule have shown mixed findings in depression²⁻⁴. Other targets highlighted the mesolimbic dopaminergic system with a small positive randomized trial of the medial forebrain bundle and small pilot trials of the lateral habenula⁷⁻¹². The absence of objective methods to assess interoceptive subjective depression-related symptoms further complicates treatment evaluation. The variability in treatment outcomes in the randomized trials hinders the establishment of DBS as a standard treatment for depression despite its urgent demand, highlighting the importance of

identifying reliable biomarkers of treatment response towards personalized and precision therapeutics.

The bed nucleus of the stria terminalis (BNST), part of the extended amygdala, is implicated in fear processing, assigning valence and social interactions¹³⁻¹⁹, which are important constructs underlying depression. The anterior BNST overlaps with the posterior nucleus accumbens with current spread from nucleus accumbens/ventral capsule targets possibly influencing BNST. The BNST has also shown preliminary efficacy as a DBS target for MDD²⁰⁻²² and OCD²³ and has been suggested to be relevant to improvements in depression²⁴. This includes a pilot open-label study suggesting potential efficacy, although a follow-up small RCT (n = 8) did not show efficacy, likely due to the small sample size, and lack of individualized optimization of stimulation parameters with an initial fixed cross-over RCT of contacts at low and moderate intensity and stimulation of the lower two and top two contacts^{21,25}. The BNST is well-placed to play an integrative role in emotional processing and regulation with extensive connections with hypothalamic, autonomic, and midbrain neurotransmitter structures¹⁴. ”

This expanded introduction provides historical context and positions the BNST within the evolving understanding of DBS targets for depression. We hope this revision addresses your concern.

Q2: Overall, this study seems like a blend of results of the clinical trial (which the authors state has been submitted for publication elsewhere) and the biomarker discovery analysis (which is the main focus of the study). Some of the results presented, such as the outcomes, the many many correlations across baseline, 3mo, 6mo, 12mo scales (in Fig. 2), and the QoL analyses seem better suited for the main clinical trial results, unless the authors can more directly tie these clinical outcome analyses to the present biomarker study.

A2: Thank you for your insightful feedback. The primary focus of this study is biomarker identification. We emphasize here that biomarkers are only relevant if they predict relevant outcomes to DBS. Note that the clinical trial focuses on the randomized controlled trial data. This data is specific to the open label trial and does not include data from the randomized trial which is reported elsewhere. We further emphasize that predicting quality-of-life (QoL) improvements and disability outcomes including the open label arm represents a crucial yet underexplored area in clinical research, as it directly reflects patients' real-world functional improvements. In our study, we specifically investigated whether physiological theta biomarkers and our neuropsychological biomarker could predict these critical clinically relevant outcomes (*See line 258-293*). Notably, we found that our identified baseline BNST theta power predicted QoL improvements at 1 year, while negative emotional bias, our neuropsychological biomarker, was associated with disability outcomes. We believe these findings bridge the gap between physiological and psychological metrics and real-world patient benefits, underscoring the translational significance of our results.

Q3: Although the authors note several times that their results are cross-validated, these methods are not entirely explained. The two instances of “cross-validation” (in my understanding) are:

(1) eyes-open versus eyes-closed -- this comparison studies state-specific effects, which is important and I commend the authors for thinking about this, but not really cross-validation because the data are still from the same groups of patients (except 2 patients excluded from eyes-closed). I suggest this should instead be framed as a method to test for the robustness of the biomarker under different state conditions, and the results should be clearly stated what was similar vs different across states.

(2) cross-validation in ridge regression – the authors state they performed 3-fold cross-validation here, and this is great but the methods do not explain details of this process (i.e., Were they 3 different groups or was there overlap in iterations?

How was performance assessed and reported? Was the model trained on 3mo data?). Although their results appear to hold up to this cross-validation, I caution the authors to claim their biomarker is generalizable given that it is tested here in a limited single-center RCT cohort and should be validated in independent dataset(s) to truly test its generalizability.

A3: Thank you for your insightful comments. We have carefully revised and clarified our methods to address your concerns.

(1) eyes-open versus eyes-closed -- We acknowledge that this comparison is not cross-validation but rather a test of biomarker robustness across different state conditions. The relevant sections have been updated to reflect this distinction, stating that the data came from the same group of patients (except for two exclusions) and emphasizing differences between states rather than datasets. The revised text now reads:

Abstract: “confirmed across eyes-open and eyes-closed states using machine learning”

Line 175-176: “Using the eyes-open dataset, we verified robustness of the biomarker through generalizability across both states.”

Line78-79: “We then confirmed with machine learning and cross-validated across eyes-open and eyes-closed states”

To clearly highlight similarities and differences across states, we have explicitly stated: *“Our findings successfully replicated the predictive value of BNST theta power but not coherence in the eyes-open dataset”.*

(2) cross-validation in ridge regression – We apologize for the missing details and have now clarified the process. The revised text provides further details of the model as follows:

“We fit spectral power features of regions or functional connectivity between regions (spectral power: 2 regions × 5 bands; coherence: 5 bands; total 15 features) into a cross-validated penalized ridge regression model with permutation feature selection to identify

the electrophysiological features predictive of clinical improvements. The modelling pipeline consisted of built-in 3-fold (without overlap) cross-validation for model hyperparameter optimization and repeated 10000 iterations for stability and better performance. Permutation feature importance was then computed for the model based on the decrease in a model score when a single feature value is randomly shuffled. The model was trained on 3-month improvement data and applied to 6- and 12-month outcomes, with R2 scores reported for model performance. Due to the small sample size, these R2 scores were not cross-validated."

We fully appreciate your caution regarding the generalizability of our biomarker. To address this, we have updated our manuscript to clarify that our results, while promising, should be viewed with caution regarding generalizability. We have explicitly removed any claims about the biomarker's generalizability and emphasized the need for further validation in independent cohorts.

Q4: Along these lines with the ridge regression, the rationale for the statistical approach presented in Fig. 3 is not clear. It appears the authors (a) performed univariate correlations to test relationships between spectral power in various frequency bands and outcomes (is this % improvement?), then (b) performed cross-validated ridge regression (reporting good R2 values but are these the cross-validated values?), and then (c) performed linear regressions of only theta power and coherence over all time points. Why not use the ridge regression model from the 3mo outcomes and try to apply it to 6mo and 12mo?

A4: Thank you for your thoughtful suggestion. First, we conducted univariate correlations between spectral power across various frequency bands and % improvement, which has now been explicitly clarified in both the methods and figures. We have clearly stated that the outcome is % improvement, calculated as [(pre-post) / pre]. See following text:

"To achieve this, we identified spectral features and functional connectivity within the BNST, NAc, hypothalamus, and prefrontal cortex that correlated with improvements [(pre-post)/pre] in depression (HAM-D, MADRS), anxiety (HAM-A) and anhedonia (DARS) at 3 months (n = 17)."

We agree that it would be valuable to apply the 3-month outcome ridge regression model to the 6-month and 12-month outcomes, and we have done so. The model was trained on the 3-month improvement data and then applied to predict the 6-month and 12-month improvements, with R2 scores reported for model performance.

However, we remain cautious about directly using the ridge regression model for prediction, as the R2 values reported are not cross-validated, and the model was primarily used for feature selection. Given the small sample size, we opted to use a simpler linear regression model for prediction in these instances. We believe this approach is more appropriate for our current dataset and provides a more conservative estimate of model performance.

Q5: Which then brings me to a critical question about correction for multiple comparisons, especially given the vast number of comparisons and correlations performed in this study. In the methods, the authors state they used FDR where appropriate. It appears to me that many of the p-values are uncorrected, but often key information is missing. Throughout the results, the test statistics and p-values should be reported (e.g., no test statistics in lines 108-111) and it should be very clear 1) whether correction for multiple comparisons was performed and 2) if the reported results survived correction for multiple comparisons.

A5: Thank you for your comment regarding the correction for multiple comparisons and the clarity of reporting test statistics and p-values. We appreciate the importance of ensuring transparency and rigor in statistical reporting. In the manuscript, we clearly indicate whether the results presented have survived FDR correction or are unadjusted p-values. For all statistical tests, we either provide the test statistics directly in the text or present them in the figures/tables. For example, in line 134-138, 266-285 or 209-225:

“Stimulation resulted in significant reductions in HAMD scores by an average of 10.1 points (SD 7.1, 95% CI 7.2-13.0, $t(25) = 7.26$, FDR adjusted $p < 0.001$), MADRS scores by 13.5 points (SD 7.8, 95% CI 10.4-16.6, $t(25) = 8.86$, FDR adjusted $p < 0.001$), and HAMA scores by 10.4 points (SD 7.2, 95% CI 7.5-13.3, $t(25) = 7.33$, FDR adjusted $p < 0.001$).”

“We evaluated whether these symptom improvements (HAMD, MADRS, HAMA, DARS) contributed to overall gains in quality-of-life and psychosocial function, applying FDR correction to control for multiple comparisons across these measurements. WHOQOL-BREF improvements were associated with greater anhedonia improvements at 6 months (DARS: Spearman’s $r = 0.54$, $p = 0.005$, survived after FDR correction) and with greater improvements in depression, anxiety, and anhedonia at 1 year (HAMD: Spearman’s $r = -0.47$, $p = 0.034$; MADRS: Spearman’s $r = -0.48$, $p = 0.029$; HAMA: Spearman’s $r = -0.52$, $p = 0.016$; DARS: Spearman’s $r = 0.46$, $p = 0.036$; all survived after FDR correction). SF-6 improvements were linked to greater anxiety and anhedonia improvements at 3 (HAMA: Spearman’s $r = -0.44$, $p = 0.025$; DARS: Spearman’s $r = 0.50$, $p = 0.009$; all survived after FDR correction) and anhedonia improvements at 6 months (DARS: Spearman’s $r = 0.57$, $p = 0.003$; survived after FDR correction). SDS improvements correlated with greater anxiety and depression improvements at 3 (HAMA: Spearman’s $r = 0.62$, $p < 0.001$; HAMD: Spearman’s $r = 0.48$, $p = 0.013$; MADRS: Spearman’s $r = 0.64$, $p < 0.001$; all survived after FDR correction) and 6 (HAMA: Spearman’s $r = 0.52$, $p = 0.007$; HAMD: Spearman’s $r = 0.53$, $p = 0.005$; MADRS: Spearman’s $r = 0.64$, $p < 0.001$; all survived after FDR correction) months and improvements in depression, anxiety, and anhedonia at 1 year (HAMA: Spearman’s $r = 0.71$, $p < 0.001$; HAMD: Spearman’s $r = 0.57$, $p = 0.007$; MADRS: Spearman’s $r = 0.67$, $p < 0.001$; DARS: Spearman’s $r = -0.44$, $p = 0.047$; all survived after FDR correction)”

“To assess the anatomical specificity of prefrontal EEG channels and their alignment with tractography, we conducted an exploratory analysis between EEG measures (spectral power and EEG-BNST coherence) from each channel and improvements in depression (HAMD, MADRS), anxiety (HAMA), and anhedonia (DARS) at 3 months and controlled for multiple comparisons using FDR correction across 7 EEG channels. Notably, only theta power in FP2 (HAMA: Spearman’s $r = -0.61$, FDR adjusted $p = 0.034$; MADRS: Spearman’s r

= -0.59, FDR adjusted $p = 0.046$) and F8 (HAMA: Spearman's $r = -0.66$, FDR adjusted $p = 0.026$; MADRS: Spearman's $r = -0.59$, FDR adjusted $p = 0.046$) showed significant associations with 3-month HAMA and MADRS improvements. Additionally, theta power across all prefrontal EEG channels was significantly correlated with 3-month HAMD improvements: FP1 (Spearman's $r = -0.59$, FDR adjusted $p = 0.025$), FP2 (Spearman's $r = -0.66$, FDR adjusted $p = 0.025$), Fz (Spearman's $r = -0.56$, FDR adjusted $p = 0.025$), F3 (Spearman's $r = -0.53$, FDR adjusted $p = 0.03$), F4 (Spearman's $r = -0.57$, FDR adjusted $p = 0.025$), F7 (Spearman's $r = -0.55$, FDR adjusted $p = 0.025$), and F8 (Spearman's $r = -0.62$, FDR adjusted $p = 0.025$)”

For analyses where unadjusted p-values are reported, we explicitly state that these are exploratory findings. For example, in line 212-215,

“On an exploratory basis, we found a state-related baseline behavioral measure, negative emotional arousal bias ((positive-negative)/neutral)), which was correlated with BNST theta power (Spearman's $r = -0.52$, unadjusted $p = 0.037$) and coherence (Spearman's $r = -0.53$, unadjusted $p = 0.029$) (Fig. 4c).”

This approach ensures clarity about which findings have undergone correction for multiple comparisons and which are exploratory in nature, providing readers with the necessary context to interpret the results accurately. We hope this addresses your concerns effectively.

Q6: The physiology-guided tractography, although very interesting and an important anatomical component, is not sufficiently explained and thus difficult to interpret. The VTA model estimates the effect of DBS on surrounding brain tissue and requires stimulation parameters as input. Yet, to my understanding, all neurophysiological recordings were performed off-stimulation. What stimulation parameters were used here? And what neurophysiological variables were assigned to each VTA for the analysis? Further details of these methods, as well as a more accurate description of exactly what the results represent are needed. If there is a disconnect between stimulation and physiology, this needs to be described and discussed as a limitation. Additionally, was modulation of the identified pathways associated with better depression/anxiety improvement in this cohort (briefly mentioned in another cohort I think in lines 283-285 but does it also apply here?). This would greatly strengthen the results. Along these lines, were there no tracts associated with worse outcomes? And did theta power also vary depending on contact location relative to the BNST or other structures?

A6: We very much appreciate your comments and very kind suggestions.

In this exploratory analysis, we aimed to identify the tracts activated during optimal stimulation (after 3 months open label parameter adjustments tailored to patients' preference) that might be associated with BNST theta recorded at the perioperative baseline. The stimulation parameters used parameters at 3-month and neurophysiological variables used perioperative baseline theta power in the BNST. The identified tracts likely represent therapeutic circuits modulated by DBS, with baseline BNST theta acting as a predictor of target engagement. The identified tracts

may be part of the mechanism underlying the treatment's effectiveness by modulating theta activity in the BNST. We have revised the relevant part as follows for clarity, in line 739-747:

“On an exploratory basis, we sought to link this baseline perioperative physiology of BNST theta and EEG-BNST theta coherence which was clinically predictive of therapeutic outcome with fiber tracts activated at 3-month stimulation (following open-label parameter adjustments tailored to patient clinical outcomes, mean [SD] amplitude: 4.45 [0.4] V) that were associated with perioperative baseline BNST theta and EEG-BNST theta coherence. We speculated that these associated fiber tracts may represent connectivity pathways relevant to the identified physiology and therapeutic effect of DBS, with perioperative baseline BNST theta or EEG-BNST theta coherence acting as a potential predictor of target engagement.”

“We applied the DBS fiber filtering approach as introduced by Baldermann et al.’s²⁶. Specifically, the E-Field was estimated using a finite element method on a four-compartment mesh describing local grey and white matter, as well as electrode contact and insulating material. Subsequently, voxel-wised normative structural connectivity seeding from bilateral E-fields were estimated using normative data sets from the Human Connectome Project at Massachusetts General Hospital (32 subjects, multi-shell diffusion-weighted imaging data).”

E-field values were used as weights to construct structural connectivity profiles. For each patient, fibers passing through a non-zero voxel of the E-field were extracted from the normative connectome and mapped onto a standardized voxelized volume with 2mm resolution. Each fiber received the weight of the maximal E-field magnitude of its passage and fiber densities were weighted by these values.”

Each fiber was assigned an R-value based on the Spearman correlation between its weighting and baseline BNST theta or EEG-BNST theta coherence. A high R-value indicated that the baseline physiology is associated with the engagement of the tract. Significant (unadjusted) tracts were then integrated to refine the physiological correlates.”

We acknowledge that we have not analyzed the effect of stimulation on physiology but rather were asking a different question of how baseline off-stimulation physiology which appears to have a clinically predictive therapeutic effect might be linked to stimulated fiber tracts by modelling the VTA of stimulation sites at 3 months and have explicitly discuss the limitation in the manuscript. See the revised text in the limitation, in line 426-429:

“We note that although we used on-stimulation E-field model estimation to link perioperative baseline physiology and fiber tracts, we did not examine the effects of stimulation on physiology thus limiting our capacity to fully assess physiology-guided tractography.”

We have incorporated a supplementary figure (**Extended Data Fig. 5a &b**) combining physiology-guided tractography with tracts directly associated with improved depression (HAMD) and anxiety (HAMA) improvements in this cohort.

Additionally, we have included sour tracts linked to worse outcomes, providing a more comprehensive analysis. Interestingly, pathways associated with better depression outcomes showed substantial overlap with physiology-guided tractography, whereas tracts linked to better anxiety outcomes were primarily found in subcortical regions and were relatively sparse. See the revised text in the limitation, in line 203-208:

“Furthermore, we identified overlapping fiber tracts associated with improved depression (HAMD) and anxiety (HAMA) outcomes. Pathways linked to greater depression improvement showed substantial overlap with physiology-guided tractography, whereas tracts associated with anxiety improvement were relatively sparse and primarily localized to subcortical regions”

Finally, we investigated whether there was any difference in theta power across the NAc, BNST, and hypothalamus. However, no significant differences were observed between these regions ($\chi^2(3) = 4.45, p = 0.217$). This lack of difference may be attributed to the distributed nature of low-frequency activity and the characteristic abnormalities exhibited by interconnected brain regions in pathological states.

Q7: Given the tractography results, I am curious if the authors investigated if their EEG results aligned with the anatomical localization of these tracts. If I understand correctly, EEG signals were aggregated across prefrontal channels, but was there any anatomical specificity for specific EEG channels observed for the prefrontal-BNST theta coherence?

A7: Thank you for this insightful question. We appreciate the opportunity to clarify and expand on our findings.

We acknowledge the limitations of interpreting data from individual EEG channels due to factors such as low signal-to-noise ratio, variability in anatomical localization across individuals, and volume conduction effects.

However, we agree this is an important area of inquiry. We conducted additional analyses to explore whether distinct prefrontal EEG channels exhibit anatomical specificity, guided by tractography findings. See the added text:

“To assess the anatomical specificity of prefrontal EEG channels and their alignment with tractography, we conducted an exploratory analysis between EEG measures (spectral power and EEG-BNST coherence) from each channel and improvements in depression (HAMD, MADRS), anxiety (HAMA), and anhedonia (DARS) at 3 months. Notably, only theta power in FP2 (HAMA: Spearman’s $r = -0.61$, FDR adjusted $p = 0.034$; MADRS: Spearman’s $r = -0.59$, FDR adjusted $p = 0.046$) and F8 (HAMA: Spearman’s $r = -0.66$, FDR adjusted $p = 0.026$; MADRS: Spearman’s $r = -0.59$, FDR adjusted $p = 0.046$) showed significant associations with 3-month HAMA and MADRS improvements. Additionally, theta power across all prefrontal EEG channels was significantly correlated with 3-month HAMD improvements: FP1 (Spearman’s $r = -0.59$, FDR adjusted $p = 0.025$), FP2 (Spearman’s $r = -0.66$, FDR adjusted $p = 0.025$), Fz (Spearman’s $r = -0.56$, FDR adjusted $p = 0.025$), F3 (Spearman’s $r = -0.53$, FDR adjusted $p = 0.03$), F4 (Spearman’s $r = -0.57$, FDR adjusted $p =$

0.025), F7 (Spearman's $r = -0.55$, FDR adjusted $p = 0.025$), and F8 (Spearman's $r = -0.62$, FDR adjusted $p = 0.025$)(Extended Data Fig. 1c)."

Interestingly, the associations for FP2 and F8 align with recent clinical interest in targeting these sites for TMS treatment in depression, both of which have demonstrated clinical efficacy and are consistent with the anatomical localization observed in tractography findings.

However, coherence between BNST and the seven EEG channels did not show significant associations with 3-month clinical improvements ($p > 0.05$). This may be due to the greater sensitivity of coherence measures to random noise within individual EEG channels.

We appreciate the reviewer's thoughtful question and believe this additional analysis offers valuable insights into the relationship between EEG findings and tractography results.

Q8: Throughout the methods and results, the authors tend to use somewhat vague terms for variables assessed (e.g., "efficacy for depression" (line 127). While I understand wanting to keep it simple, it is also important to explain exactly what is being assessed (e.g., change in HAMD and MADRS scores at 3 months compared to preoperative baseline). At times it is difficult to tell whether the analysis focused on scores at baseline vs improvement scores at follow-up.

A8: Thank you for your constructive feedback. In response, we have made several adjustments throughout the manuscript to enhance clarity and precision.

Specifically, we have revised all vague terms in Figures to explicitly differentiate between baseline and improvement scores. Additionally, we have revised key sections of the manuscript for better clarity. These revisions now clearly distinguish between baseline scores and improvements at follow-up, addressing the concerns raised and improving the overall clarity of the manuscript.

For example:

1. Original: *"To achieve this, we identified spectral features and functional connectivity within the BNST, NAc, hypothalamus, and prefrontal cortex that correlated with DBS efficacy for depression (HAMD, MADRS), anxiety (HAMA) and anhedonia (DARS) at 3 months ($n = 17$)."*

Revised: *"We identified spectral features and functional connectivity within the BNST, NAc, hypothalamus, and prefrontal cortex that correlated with improvements [defined as (pre-post)/pre] in depression (HAMD, MADRS), anxiety (HAMA), and anhedonia (DARS) at 3 months relative to baseline ($n = 17$)."*

2. Original: *"We evaluated whether these improvements contributed to overall gains in quality-of-life and psychosocial function. WHOQOL-BREF measurements were associated with better anhedonia outcomes at 3 and 6 months and with better*

depression, anxiety, and anhedonia outcomes at 1 year. SF-6 measurements were linked to better anxiety and anhedonia outcomes at 3 and 6 months, and better anxiety outcomes at 1 year. SDS measurements correlated with better anxiety and depression outcomes at 3 and 6 months, and improvements in depression, anxiety, and anhedonia at 1 year”

Revised: “We assessed whether these symptom improvements (HAMD, MADRS, HAMA, DARS) contributed to overall quality-of-life and psychosocial gains. WHOQOL-BREF improvements were linked to greater anhedonia improvements at 3 and 6 months and to greater improvements in depression, anxiety, and anhedonia at 1 year. SF-6 improvements were associated with greater anxiety and anhedonia improvements at 3 and 6 months and with anxiety improvements at 1 year. SDS improvements correlated with greater anxiety and depression improvements at 3 and 6 months and with improvements in depression, anxiety, and anhedonia at 1 year.”

Q9: The discussion lacks depth in literature review and would benefit from some expanded discussion on a few points:

(1) Further discussion comparing the present results to previous studies that have investigated biomarkers in DBS for depression would help situate the study’s findings within the broader context of the field.

(2) The authors found a very interesting result that their biomarker was linked to anxiety and depression but not anhedonia. Some additional discussion about how/why these biomarkers may be “subtype-specific” and potential implications would be beneficial.

(3) The authors very briefly allude to the potential clinical utility of their findings. More discussion about how these biomarkers could be incorporated into patient-selection, real-time DBS programming, or even closed-loop systems would enhance the clinical relevance of the study. The noninvasive stimulation angle is interesting given the fiber pathways identified – what would be the proposed target and how would the present results inform future protocols?

A9: Thank you very much for your valuable suggestions. We greatly appreciate your thoughtful comments, which have been instrumental in enhancing the depth of our discussion. We have carefully considered each point and expanded our responses to provide a more comprehensive understanding of our findings in the context of the broader field.

1. Comparison with previous studies investigated biomarkers in DBS for depression:

We have now incorporated a more detailed discussion comparing our results with prior research, emphasizing how our study builds upon and extends the current understanding of DBS biomarkers in the context of depression. In response, we have added the following discussion:

“To date, advances in MDD DBS studies have introduced behavioral and biometric (e.g., facial expression, speech, heart rate or heart rate variability), neuroimaging (e.g., tractography), and neurophysiological markers (e.g., SCC beta power, aperiodic 1/f activity) to inform symptom severity, treatment response and remission^{1,27-31}. Multiple crucial advances have informed biomarker identification in MDD DBS. Recent studies emphasize a key role for precision targeting and individual tractography of the convergence of critical pathways within the SCC and the integrity of the cingulum bundle^{32,33}. Critical studies highlight physiological predictive biomarkers to separate out the disease illness state versus the well state implicating lateralized beta and gamma power with relevance for tracking relapse²⁷. Personalized approaches through multisource stereotactic EEG recordings highlight a role for recording amygdala reactivity and stimulation through ventral capsule connecting fibers^{34,35}. Intracranial recordings have also identified alpha desynchronization related to emotional imagery in the subthalamic nucleus and habenula which shift emotional valence bias with time-locked alpha specific stimulation^{36,37}. While these biomarkers hold strong translational potential, the present study advances the field in three key areas: First, it integrates multimodal approaches to improve the robustness and clarity of biomarker identification. Second, it identifies circuit-specific biomarkers in the BNST and prefrontal cortex linked to anxiety biotypes, aiding in depression subtype differentiation, personalized treatment decision, and targeted symptom monitoring. Third, it establishes these biomarkers through cutting-edge longitudinal LFP recordings as tracking transient mood states in MDD patients, further validating their role in guiding DBS therapeutic responses and potential as a biomarker for closed loop stimulation.”

2. Biomarker Subtype Specificity

Thank you for your thoughtful comment and for highlighting the interesting result regarding the link between our biomarker and anxiety/depression, but not anhedonia. We appreciate your suggestion to expand on the potential reasons why these biomarkers may be “subtype-specific.”

In response, we have added the following discussion:

“Interestingly, the BNST theta biomarker and psychological marker may be anxiety associated biotype-specific. Evidence for the role of BNST in anxiety is robust, with BNST activity underlying unpredictable threat, social stress, contextual fear, and startle responses characteristic of anxiety and fear disorders³⁸. BNST DBS has also shown efficacy in treating OCD and depression with high comorbidity with anxiety symptoms^{21,23,39}. Previous research suggest ventral internal capsule DBS may be effective for depression with anhedonia, while subcallosal cingulate DBS may better address depression characterized by sadness or mental anguish⁴⁰. These findings underscore the importance of distinguishing depression biotypes addressing the heterogeneity of depression and personalized DBS targets, circuits, and stimulation parameters to individual phenotypes or biomarkers.”

3. Clinical Utility:

We appreciate your recommendation to further discuss how these biomarkers could be applied in patient selection, real-time DBS programming, and closed-loop systems. We have added the following discussion to address your suggestion:

“Our findings suggest a preoperative EEG physiological and psychological negative emotional bias prognostic baseline biomarker with the potential to help guide patient stratification and DBS target selection for subgroups sensitive to BNST-NAc DBS. Subsequent intracranial and EEG physiology might help guide stimulation optimization. Further work integrating multimodal predictors is recommended to better predict both symptom and psychosocial functional outcomes. Given the capacity of BNST theta to track inter-individual mood states, this may be a promising biomarker, aligning with the growing interest in closed-loop therapeutic strategies for neuropsychiatric disorders. Our study also has potential implications for non-invasive stimulation protocols. It sheds light on the antidepressant benefit arising from theta burst stimulation (TBS) over prefrontal areas through top-down regulation⁴¹. The identified fiber pathways are relevant to the novel repetitive transcranial magnetic stimulation (rTMS) protocols targeting the orbitofrontal cortex (OFC) as an alternative or augmentation for treatment-resistant depression⁴²⁻⁴⁴. Moreover, our findings pave the way for a personalized, circuit-selective and frequency-based approach for MDD across available multiple intervention modalities.”

Q10: The limitations section in the discussion is missing several key limitations to be mentioned:

- (1) some results were not robust to correction for multiple comparisons, and correlations only explain a fraction of the variability in outcomes and thus should be further explored in independent datasets
- (2) potential compound effects of stimulation on neural activity and neuroplasticity – we may not know the ideal washout period and thus cannot rule out carryover effects across active/sham
- (3) the VTA model requires many assumptions about the mechanisms of DBS that are unknown, and thus serves as an estimate of the effect of DBS on surrounding tissues and is not ground truth
- (4) normative connectome is a valid approach to estimate connectivity (can cite several other papers out there from Horn et al and others using this approach) but this may not capture individual connectivity differences, which may be important
- (5) decoupling of neuropsychological task and neural recordings – future studies could combine these together to directly evaluate their relationship
- (6) use of momentary assessments of mood/anxiety vs clinical rating scale scores – “state” vs “trait” could be worth discussing

A10: We greatly appreciate your insightful comments and fully agree with all the points raised. We have taken them into careful consideration and have addressed

each limitation in the revised manuscript. Specifically, we have expanded the discussion on the following points:

“The study is not without limitations. First, our findings in the BNST appear to be predictive of depressive outcomes associated with anxiety but not anhedonia suggesting potential differential effects on depression subtypes. Further larger studies are indicated to assess the effects on depressive subtypes.

Second, although patients were blinded to active and sham DBS conditions, they could correctly identify stimulation settings. The potential compound effects of stimulation on neural activity and neuroplasticity, coupled with the absence of a defined washout period, raise the possibility of carryover effects. Notably, BNST-theta power remained low in the sham phase for some patients who started the crossover study with stimulation, suggesting a potential carryover effect and highlighting the need for longer washout periods in future studies to mitigate these effects. Moreover, wireless recordings were conducted off-stimulation, and stimulation artifacts were difficult to adequately control, limiting our ability to accurately capture stimulation effects and potentially explaining the divergent relationship between theta and HAMD observed after active or chronic stimulation.

Third, some results were not robust to correction for multiple comparisons, and observed correlations explained limited variability, highlighting the need for validation in independent larger datasets. Fourth, E-field models are based on some assumptions about DBS mechanisms that remain uncertain, serving as approximations rather than definitive representations of its effects on surrounding tissues⁴⁵. While normative connectomes are validated for estimating connectivity, they may not capture individual differences crucial for optimizing outcomes⁴⁶⁻⁴⁸. We note that although we used on-stimulation E-field model estimation to link perioperative baseline physiology and fiber tracts, we did not examine the effects of stimulation on physiology thus limiting our capacity to fully assess physiology-guided tractography. Finally, decoupling of neuropsychological task and neural recordings limits direct insights into their relationship, highlighting the need for task-based neurophysiological studies.”

We also address the trait versus state distinction as follows:

“Additionally, BNST theta activity correlated with improvements in HAMD, MADRS, and HAMA scores but specifically tracked momentary anxiety changes. These differing measures represent the different temporal dimensions of the mood and anxiety. Clinical rating scales (e.g., HAMD, MADRS, and HAMA) assess symptoms over the past two weeks and reflect general levels of anxiety or mood, often considered pathological traits. In contrast, ecological momentary VAS ratings capture fluctuating emotional dynamics, reflecting transient states⁴⁹. While depression and anxiety severity are often highly correlated in MDD patients, the dissociation between anxiety and mood states in BNST physiopathology in this study may reflect heterogeneous symptomatology and physiopathology across timescales, with transient states introducing greater variability and context dependence⁵⁰. Notably, although momentary VAS ratings are commonly used, validated, objective, and reproducible measures for state anxiety and mood (e.g. heart rate, heart rate variability, skin resistance) are needed to substantiate the findings.”

Q11: The extended figures/tables were not included in the files for download and thus I was not able to evaluate these.

A11: We appreciate your feedback and sincerely apologize for the oversight in providing the extended figures and tables. We have now ensured that all supplementary materials, including the extended figures and tables, are uploaded and available for review. These materials provide additional context and details to support the main findings of our study. Please let us know if there are any issues accessing them, and we are happy to assist further. Thank you for bringing this to our attention.

Minor Edits/Comments

Title/Abstract/Introduction

Q12: - The title could be reworked to be clearer – the “Physiological Intracranial Prefrontal – Bed Nucleus of the Stria Terminalis” does not really make sense.

A12: We appreciate the reviewer’s suggestion regarding the clarity of our title. In response, we have revised it to: *“Predicting Therapeutic Depression Outcomes: Bridging Prefrontal–Bed Nucleus of the Stria Terminalis Physiological and Neuropsychological Biomarkers.”* This revision enhances readability while preserving the key focus of our study on the integration of physiological and neuropsychological biomarkers in predicting treatment outcomes for depression. We believe this new title more accurately conveys the scope and significance of our work.

Q13: - The abstract focuses mainly on the study’s results and implications. A brief summary of the key methodological aspects would be helpful for readers as well.

A13: We appreciate the reviewer’s suggestion to include a brief summary of key methodological aspects in the abstract. In response, we have revised the abstract to provide greater clarity on our study design while maintaining focus on the results and implications:

“Therapeutic options for refractory depression are urgently needed. Deep brain stimulation (DBS) of the bed nucleus of the stria terminalis (BNST), an extended amygdala structure, and nucleus accumbens (NAc) represents a potential therapeutic target. Here, we identified an objective intracranial physiological biomarker in 26 refractory depression patients using acute and chronic intracranial recordings, machine learning, and an integrated framework combining electrophysiology, neuroimaging, and behavior: lower BNST theta and prefrontal-BNST coherence with top-down connectivity predicted better depression outcomes and quality-of-life after chronic stimulation at 3, 6 and 12 months, confirmed using separate data sets and machine learning. We identified a physiology-guided connectivity network involved dorsal anterior cingulate and lateral inferior frontal

cortex tracts. These biomarkers, linked to negative emotional bias and anxiety, highlight the efficacy of BNST-NAc DBS for refractory depression and has potential broader clinical implications.”

Q14: - Line 33 – define DBS acronym (instead defined in sentence soon after)

A14: Thank you for your attention to detail. We have revised the text to define the DBS acronym at its first mention in line 33 for clarity.

Q15: - Line 37 – “medial prefrontal” should be medial prefrontal cortex for mPFC

A15: Thank you for your careful review. We have corrected “medial prefrontal” to “medial prefrontal cortex”.

Q16: - Line 42 – another motivation for this study (which could be added before “This variability hinders...”) is the lack of objective methods to evaluate depression-related symptoms given their interoceptive nature

A16: Thank you for the insightful suggestion. We have incorporated this point by adding:

“The absence of objective methods to assess interoceptive subjective depression-related symptoms further complicates treatment evaluation.”

Q17: - Line 47 – preliminary efficacy – should specify if evidence comes from open-label, RCTs, etc. More description of previous literature would be helpful for context

A17: Thank you for your suggestion. We have added: *“This includes a pilot open-label study suggesting potential efficacy, although a follow-up small RCT (n = 8) did not show efficacy, likely due to the small sample size, and lack of individualized optimization of stimulation parameters with an initial fixed cross-over RCT of contacts at low and moderate intensity and stimulation of the lower two and top two contacts”*. This revision provides more context on the nature of prior evidence and clarifies the limitations of previous studies.

Q18: - Line 48-50 – “Compelling target rationale...” – this sentence is a bit awkwardly worded. Could modify to something like “The rationale for BNST as a DBS target is largely based on its involvement in features of depression, including sustained fear, stress responses, social behavior, and valence surveillance.”

A18: Thank you for the helpful suggestion. We have revised the sentence accordingly.

Q19: - Line 51-52 – missing comma between hypothalamic and autonomic

A19: Thank you for catching that. We have corrected the missing comma between hypothalamic and autonomic to ensure proper punctuation.

Results

Q20: - Line 53 – typo – “relative” should be “relatively”

A20: Thank you for pointing that out. We have corrected the typo, changing “relative” to “relatively”, for grammatical accuracy.

Q21: - Line 74 – why is the cohort so predominantly male?

A21: Thank you for your observation. We acknowledge the predominance of male participants in the cohort. Recruitment was conducted without any gender-specific criteria or bias. This gender distribution likely reflects the demographics of individuals seeking treatment at the neurosurgical center during the study period. One possible explanation is that males may be more inclined to accept invasive procedures or treatments perceived as higher risk similar to observations in the Parkinson’s disease DBS field. We recognize the importance of gender diversity and will explore strategies to ensure a more balanced representation in future studies.

Q22: - Line 87-90 – were postoperative wireless recordings performed in-clinic?

A22: Thank you for your question. Yes, postoperative wireless recordings were performed in-clinic. We have clarified this in the Methods section: *“Ten patients participated in the daily in-lab electrophysiological recordings and momentary self-report behavioral assessments during the randomized crossover phase.”* Additionally, we explicitly state in the main text: *“All postoperative wireless recordings were performed in-clinic.”*

Q23: - Line 91 and Line 98 – what is meant by “baseline” – preoperatively? If so, how soon before surgery?

A23: Lines 91 and 98: By “baseline,” we mean preoperative assessment. We have clarified this as: *“Throughout the study, patients were assessed at baseline (1–3 days pre-surgery) and then monthly post-surgery using standardized measures for depression.”*

Q24: - Line 92-93 – the rationale for using both the HAMD-17 and MADRS as measures for depression should be explained – how do they complement each other and why do they both need to be included in all analyses?

A24: We included both the Hamilton Depression Rating Scale (HAMD-17) and the Montgomery–Åsberg Depression Rating Scale (MADRS) to provide a comprehensive assessment of depressive symptoms. The HAMD-17, developed for patients with major depression, evaluates depressive, anxious, and somatic symptoms and is considered multidimensional. The MADRS, a unifactorial measure more sensitive to treatment changes, was used for complementary evaluation. Using both scales allowed us to capture a broader spectrum of depressive symptomatology, thereby enhancing the reliability and validity of our findings. Consequently, we revised the Clinical Assessments section in the Methods as follows:

“Depression symptoms were assessed using the HAMD-17, with scores ranging from 0 to 52, where higher scores indicated more severe symptoms⁵¹. A patient was considered a responder if they exhibited a decrease of at least 50% in their HAMD score compared to baseline. Remission was defined as achieving a HAMD-17 score of less than 8. The HAMD-17 developed for patients with major depression assesses depressive, anxious and somatic symptoms and is considered multidimensional⁵². Montgomery-Asberg Depression Rating Scale (MADRS), a unifactorial measure assessing primary sadness or lassitude and more sensitive to treatment changes, was used for complementary depression evaluation⁵³. The MADRS ranges from 0 to 60, with higher scores indicating more severe symptoms⁵⁴. ”

Q25: - Line 101-103 – is this is during the open-label phase? Please clarify.

A25: Thank you for the clarification request. Yes, this section refers to the open-label phase. We have updated the text to include: *“During the open-label phase.”* to make this clear for the reader.

Q26: - Line 103-104 – was multisite monopolar stimulation part of the study design, or was this the result of the programming process? Please clarify.

A26: Thank you for the thoughtful inquiry. Multisite monopolar stimulation was determined based on programming tailored to each patient’s needs. The detailed programming procedure is outlined in the Surgical Procedure and Stimulation section of the Methods. To clarify, we have revised the corresponding sentence as follows: *“At the last follow-up (19 ± 8.5 months), all patients received multi-site monopolar stimulation, customized through programming to meet individual needs”.*

Q27: - Line 115-117 – “Notably at a cohort level...” – this sentence is confusing. I think it would be clearer to say something like “At preoperative baseline, anxiety and depression were highly related (as shown by high correlations between HAMD, MADRS, and HAMA scores), while anhedonia was not correlated with depression or anxiety scores, indicating potentially different clinical phenotypes.”

A27: Thank you for your suggestion. We have revised the sentence as suggested.

Q28: - Line 115-116 – I’m also unsure whether correlations between depression/anxiety/anhedonia scores are a valid way of discerning phenotypes of response to DBS. Wouldn’t it be clearer to assess the effect of DBS on these symptoms separately? Would disappearance of correlation between two symptoms really indicate a different response?

A28: Thank you for your insightful comment. We acknowledge the limitations of using correlations between depression, anxiety, and anhedonia scores to infer distinct DBS response phenotypes. While symptom decoupling may reflect differential symptom responses, heterogeneous patient trajectories, or potential confounding factors, we have removed any inference suggesting it represents a distinct response to avoid overinterpretation. The corresponding figure has been moved to the supplementary materials, allowing interested readers to examine the data without overstating its implications.

Q29: - Line 127-128 – is “efficacy for depression” defined as change in these scores compare to baseline?

A29: Thank you for your question. Yes, efficacy for depression is defined as the change in these scores compared to baseline. To improve clarity, we have revised the text to:

“To achieve this, we identified spectral features and functional connectivity within the BNST, NAc, hypothalamus, and prefrontal cortex that correlated with improvements [(pre-post)/pre] in depression (HAMD, MADRS), anxiety (HAMA) and anhedonia (DARS) at 3 months (n = 17)”.

Q30: - Line 132 – “we only found predictive effects and no state related effects” – please clarify, does this mean the features were not different between eyes open vs eyes closed?

A30: Apologies for the vague sentence. We aimed to emphasize that our biomarker is predictive rather than related to the disease state. To clarify, we have revised the text as follows:

“Notably, we found only predictive effects and no disease state-related effects associated with baseline symptom severity.”

Q31: - Line 133 – unclear what “5-frequency band average power” means?

A31: Thank you for pointing that out. We agree that “5-frequency band average power” could be unclear, so we have reverted to using “*spectral features*” for clarity and consistency throughout the manuscript.

Q32: - Line 146-149 – specifying that the model was trained on eyes-open vs eyes-closed would be helpful. Also, throughout manuscript these should be written as “eyes-open” or “eyes-closed”

A32: Thank you for the suggestion. We have clarified that the model was trained on eyes-open vs. eyes-closed conditions: *“Critically, BNST theta power and prefrontal-BNST theta coherence were the two most informative features for the training model in the eyes-closed dataset”*. Additionally, we have made sure to use “eyes-open” and “eyes-closed” consistently throughout the manuscript.

Q33: - Line 155 – “predicted” – be careful, this is not prediction because not validated. Should use “correlated with” instead

A33: We appreciate this cautious wording and acknowledge this prediction is not validated. We corrected the relevant part as suggested, for example:

“Top-down theta band Granger causality at baseline correlated with theta power (prefrontal EEG: Spearman’s $r = 0.51$, FDR adjusted $p = 0.04$; BNST: Spearman’s $r = 0.72$, FDR adjusted $p = 0.006$) and coherence (Spearman’s $r = 0.58$, FDR adjusted $p = 0.026$) and also correlated with depression and anxiety 3-month improvements (HAMA: Spearman’s $r = -0.53$, $p = 0.028$; MADRS: Spearman’s $r = -0.52$, $p = 0.031$; HAMD: Spearman’s $r = -0.53$, $p = 0.027$; all survived after FDR correction).”

Q34: - Line 179 – what does “fluctuations in BNST theta-band activity” mean? Do you mean that BNST theta-band activity correlated with HAMD scores?

A34: Thank you for the clarification. Yes, “fluctuations in BNST theta-band activity” refers to the correlation between BNST theta-band activity and HAMD scores. To make this clearer, we revised the sentence to: *“We showed that BNST theta-band activity correlated significantly with HAMD scores (Spearman’s $r = 0.42$, $p = 0.027$).”* This revision provides a more direct and understandable explanation.

Q35: - Line 181 – was the amount of reduction in theta power correlated with the amount of improvement in depression symptoms? Also please clarify, was theta power reduced compared to baseline off stim recording?

A35: Thank you for this insightful question. Although we attempted to correlate the reduction in theta power with the improvement in depression symptoms, the result was not statistically significant. This may be attributed to the relatively small sample size, where outliers can heavily influence outcomes, and individual differences in the physiological thresholds required to elicit behavioral changes.

Additionally, we confirm that theta power was reduced compared to the baseline off-stimulation state. We have clarified this in the manuscript as follows: *“Furthermore, active BNST-NAc DBS significantly reduced BNST theta-band activity from the baseline off-stimulation state compared to sham stimulation.”*

Q36: - Line 187-190 – this analysis and rationale is a bit confusing. This comes across like a strange roundabout way of saying theta power was correlated with both anxiety and mood, even though it was only correlated with anxiety. Some discussion about momentary assessments vs classic MADRS/HAMD may also be warranted.

A36: Thank you for your comments. We recognize that our explanation may seem indirect. However, mood and anxiety can be persistent, episodic, or both. Momentary assessments of anxiety provide distinct temporal and pathological insights compared to classic MADRS/HAMD scores, which reflect an individual’s general anxiety level over a longer period. In clinical scores, depression and anxiety severity are often highly correlated and difficult to dissociate in MDD patients. While BNST theta cannot separate the predictive effects in clinical scores, it differentiates momentary anxiety and mood states, with greater sensitivity to anxiety.

We have also included a thorough discussion on momentary assessments versus classic MADRS/HAMD in the discussion section, as mentioned above.

Q37: - Line 203-206 – it needs to be clearer that theta power and coherence were not recorded during the task. If I understand correctly, the task preceded the recordings (baseline pre-DBS?) and thus theta may have changed due to compound effects of stimulation?

A37: Apologies for the confusion. The affective task was conducted during the perioperative period, while resting-state electrophysiological recordings were performed separately. To clarify, we added the following:

“To investigate, we assessed subjective emotional bias by evaluating emotional valence and arousal responses to visual emotional stimuli at the perioperative baseline (the physiological results of this task-based measure are reported separately).” Additionally, we noted: “On an exploratory basis, we identified a state-related baseline behavioral measure—negative emotional arousal bias ((positive-negative)/neutral)—which

correlated with resting-state BNST theta power (Spearman's $r = -0.52$, unadjusted $p = 0.037$) and coherence (Spearman's $r = -0.53$, unadjusted $p = 0.029$)."

Discussion

Q38: - Line 247 – tone down the “state independent, ecologically valid, and generalizable” descriptors unless the results can be more clearly stated. What is meant by ecologically valid and how do the data support this?

A38: Thank you for your valuable feedback. We initially considered the prediction of quality of life, disability, and longitudinal monitoring as aspects of ecological validity. However, to ensure accuracy and avoid overstatement, we recognize that true ecological validity requires real-world experiments. Therefore, we have removed this qualifier and revised the sentence as follows:

“To summarize, we identified state-independent, objective theta-band electrophysiological biomarkers within the prefrontal-BNST network predictive of depression therapeutic outcomes.”

Q39: - Line 252 – not a separate data set – rephrase to say across eyes-open and eyes-closed states

A39: Thank you for your suggestion. We have corrected the sentence as advised.

Q40: - Line 263-271 – citing oscillatory activity in DBS for PD is not critically relevant to the focus of this paper. Instead, it would be more beneficial to expand this paragraph to discuss work that relates to biomarkers of depression and DBS (see main comment above)

A40: Thank you for your helpful feedback. In response, we have removed the reference to oscillatory activity in DBS for Parkinson's disease and expanded the paragraph to focus on work related to biomarkers of depression and DBS, in line with the main focus of our paper.

Q41: - Line 276 – what compelling evidence supports the role of theta?

A41: Thank you for the comment. We revised the text for clearer classification: “Human electrophysiology and neuromodulation studies support a prominent role of theta oscillations in the PFC-limbic circuit underlying depression pathophysiology.”

Q42: - Line 287-288 – what was the rationale for comparing to the OCD tracts? And why do the authors think that these tracts are more dorsal – what might be the underlying mechanism? More discussion here to generate hypotheses would be useful.

A42: Thank you for your thoughtful feedback. We compared the identified tracts to those associated with OCD because both MDD and OCD share overlapping symptomatology and neural circuits, and both disorders may benefit from BNST DBS.

The more dorsal projection of the identified tracts in the anterior cingulate may reflect the distinct roles of dorsal and ventral pathways in regulating emotional versus compulsive processes. Specifically, dorsal pathways may be implicated in affective regulation, which is central to MDD, while ventral pathways are more involved in compulsive behavior and response inhibition, key features of OCD. These distinctions may help explain the different therapeutic outcomes observed with DBS targeting these regions in each disorder. The revised text is as follows:

“The terminus points of the identified tracts converge with optimal therapeutic outcomes identified previously from our MDD sample and from convergent DBS studies for OCD targeting the ventral internal capsule, subthalamic nucleus and BNST^{55,56}. While OCD and MDD involve overlapping symptomatology and cognitive-affective networks, the convergence of tracts across disorders and multiple surgical targets underscores the importance of symptom-specific connectivity profiles in optimizing DBS outcomes. Although we have not statistically compared our identified projections, our identified tracts appear to have a more dorsal posterior projection in the anterior cingulate than tracts identified as the optimal therapeutic outcomes for OCD DBS targeting the ventral internal capsule⁵⁶. This aligns with the dorsal-ventral gradient of cortico-striatal circuits, where dorsal pathways regulate affective processes in MDD and ventral pathways mediate response inhibition and compulsive habit formation in OCD^{57,58}.”

Q43: - Line 290-298 – this paragraph repeats results and could instead expand more on previous studies of emotional arousal bias in depression and how it fits in here, as well as neuropsychological biomarker use in the future.

A43: Thank you for the insightful feedback. We have reviewed the text and expanded the discussion as follows:

“An emotional information processing bias has been implicated in neural mechanisms underlying depression, involving networks of emotional salience and attention (e.g., amygdala, insula) and regions responsible for emotion monitoring, evaluation, and regulation (e.g., anterior cingulate, dorsolateral prefrontal cortex)⁵⁹. This bias has been shown to be a behavioral predictor of treatment efficacy in mood disorders, with a positive bias anticipating better outcomes⁶⁰. Reversing this bias has been suggested to be central to the antidepressant effects of pharmacotherapy⁶¹⁻⁶³. A preference for positive information is linked to improved emotional regulation and rephrasing capacity, thus perhaps enhancing the capacity for positive behavioral reinforcement and augmenting psychotherapeutic approaches following DBS treatment⁶⁴. This preference may also reflect

*greater neuroplasticity in depression pathology*⁶⁵. *This neuropsychological approach offers a clinically practical, sensitive method for assessing depressive states and predicting outcomes, particularly in refractory depression requiring DBS, where conventional scales lack discriminative power due to ceiling effects. Together these findings highlight a role for a prefrontal-BNST physiological network and a neuropsychological predictor of antidepressant outcomes from chronic DBS of the BNST-NAc.*"

Q44: - Line 305 – again, careful about use of “generalizable”

A44: Thank you for the comment. We have removed the word “generalizable” to ensure greater accuracy in the text.

Methods

Q45: - Line 443-444 – should “failure of adequate psychotherapy and ECT” be a separate number in this list?

A45: Thank you for your comment. To clarify, the criterion reads: *“4) lack of antidepressant response to a minimum of three antidepressant treatments of adequate dose and duration, including at least two medications from two different classes, and failure of adequate psychotherapy or electroconvulsive therapy (ECT) (either poor response, intolerance or rejection).”* The failure of psychotherapy or ECT is included as part of the broader criterion for inadequate treatment response.

Q46: - Line 460-461 – are the contacts all cylindrical or are there also segmented? Please specify

A46: Thank you for the question. All the contacts are cylindrical.

Q47: - Line 509 – missing “and” between (SF-36) and functional disability

A47: Thank you for pointing that out. We have revised the sentence to include “and” between “(SF-36)” and “functional disability”.

Q48: - Line 515 – could add citation(s) about task validity

A48: Thank you for the suggestion. We have added the appropriate citation: *“LANG, P. J., GREENWALD, M. K., BRADLEY, M. M. & HAMM, A. O. Looking at pictures: Affective, facial, visceral, and behavioral reactions. Psychophysiology 30, 261–273 (1993).”* regarding task validity to strengthen the context of the discussion.

Q49: - Line 526 – clarify what is meant by baseline – is this preoperative? This needs to be clarified throughout the manuscript as well

A49: Thank you for your comment. We confirm that “baseline” refers to the perioperative period while electrodes were externalized. We have clarified this throughout the manuscript for consistency and to avoid any ambiguity.

Q50: - Line 538 – Baldermann et al approach implemented in Lead DBS? See major comment above about this analysis and add info about stimulation and neurophysiology link and the rationale for this. What “identified neurophysiology” was used in this analysis?

A50: Thank you for your insightful comments. Our reference to the approach described by Baldermann et al. specifically pertains to the DBS fiber filtering method applied in our analysis. To clarify, we have revised the text as follows:

“We applied the DBS fiber filtering approach as introduced by Baldermann et al.’s²⁶.”

Additionally, we have expanded on the link between stimulation and neurophysiology, clarified the meaning of “identified neurophysiology,” and provided a rationale for the analysis as suggested. The revised text is as follows:

“On an exploratory basis, we sought to link this baseline perioperative physiology of BNST theta and EEG=BNST theta coherence which was clinically predictive of therapeutic outcome with fiber tracts activated at 3-month stimulation (following open-label parameter adjustments tailored to patient clinical outcomes) that were associated with perioperative baseline BNST theta and EEG-BNST theta coherence. We speculated that these associated fiber tracts may represent connectivity pathways relevant to the identified physiology and therapeutic effect of DBS, with perioperative baseline BNST theta or EEG-BNST theta coherence acting as a potential predictor of target engagement.

“We applied the DBS fiber filtering approach as introduced by Baldermann et al.’s²⁶. Specifically, the E-Field was estimated using a finite element method on a four-compartment mesh describing local grey and white matter, as well as electrode contact and insulating material. Subsequently, voxel-wised normative structural connectivity seeding from bilateral E-fields were estimated using normative data sets from the Human Connectome Project at Massachusetts General Hospital (32 subjects, multi-shell diffusion-weighted imaging data).

E-field values were used as weights to construct structural connectivity profiles. For each patient, fibers passing through a non-zero voxel of the E-field were extracted from the normative connectome and mapped onto a standardized voxelized volume with 2mm resolution. Each fiber received the weight of the maximal E-field magnitude of its passage and fiber densities were weighted by these values.

Each fiber was assigned an R-value based on the Spearman correlation between its weighting and baseline BNST theta or EEG-BNST theta coherence. A high R-value indicated that the baseline physiology is associated with the engagement of the tract. Significant (unadjusted) tracts were then integrated to refine the physiological correlates.”

We hope this revision clarifies our approach and adequately addresses the reviewer’s concerns.

Q51: - Line 556 – in the three that were unable to record, was this because their stimulation parameters were not bipolar-friendly? Was the sandwich configuration required?

A51: Apologies for the misunderstanding. These patients were excluded because none of their contacts met the BNST targeting criteria, defined as being within or <1mm from the BNST border, despite intact stimulation. The text was revised for clarity:

“Twenty-six patients underwent perioperative signal recording. Nine were excluded: three due to excessive data noise (>50%), three for incomplete 3-month follow-up, and three because contacts did not meet BNST targeting criteria for this stringent physiological analysis.”

Q52: - Line 563-564 – could provide some more rationale for eyes-open vs eyes-closed as assessing state effects

A52: Thank you for the comment. The rationale for including both eyes-open and eyes-closed conditions is to assess potential state-dependent effects on brain activity. Eyes-closed recordings primarily capture intrinsic neural processes by reducing external visual input, whereas eyes-open recordings maintain natural sensory engagement and attentional states. Comparing these conditions helps differentiate between baseline neural activity and state-driven effects, offering a more comprehensive understanding of functional dynamics. We have clarified this rationale in the revised text:

“To assess potential sensory-attentional state effects on brain activity, recordings were conducted under two conditions: eyes open and eyes closed”.

Q53: - Line 567-568 – focused only on frontal electrodes – rationale for excluding other brain regions?

A53: Thank you for the comment. Our focus on frontal electrodes was guided by both theoretical and practical considerations. Theoretically, prefrontal-BNST connectivity is central to affective processing and regulation. By targeting frontal

regions, we specifically aligned with our study's aim to explore prefrontal-BNST dynamics, which underpin the neural mechanisms of interest. Practically, postoperative constraints—such as surgical bandages over the forehead—limited electrode placement options. Moreover, restricting recordings to the frontal region minimized patient discomfort and enhanced compliance.

This has been clarified in the manuscript:

“EEG was obtained from 7 frontal electrodes (Fp1, Fp2, F3, F4, F7, F8, Fz) using the 10-20 placement system. We focused on frontal electrodes due to their critical role in affective processing and relevance to prefrontal-BNST connectivity, essential for mood and anxiety regulation. Moreover, the surgical bandages over the forehead limited electrode placement options, and restricting the number of electrodes helped minimize patient discomfort during postoperative recordings.”

Q54: - Line 584 – was wireless data streaming performed via the IPG? And were these recording done while in-clinic? Please clarify

A54: We revised the text as follows for clarity: “Wireless data streaming was performed via the IPG while in-clinic with a sample rate of 415Hz or 1000Hz (for P3).”

Q55: - Line 591 – typo “moderate state”

A55: Thank you for pointing that out. We have corrected the typo, changing “moderate state” as suggested.

Q56: - Line 599-601 – how many data epochs were omitted? Could mention use of the EOG recordings as well here for identifying blinks/saccades

A56: Thank you for your thoughtful comment. In response, we have clarified the data omission details as follows: *“For the eyes-closed datasets, $62 \pm 50s$ (range: 4-208s) were removed; for the eyes-open datasets, $48 \pm 40s$ (range: 0-141s) were excluded. The final analysis included 17 patients' eyes-closed datasets ($230 \pm 55s$, range: 140-308s) and 15 patients' eyes-open datasets ($227 \pm 52s$, range: 94-282s).”*

Additionally, we have incorporated the use of EOG recordings for identifying blinks and saccades during the Independent Component Analysis (ICA), followed by visual inspection for artifact rejection: *“Independent Component Analysis (ICA) including EOG for identifying blinks and saccades, followed by visual inspection, was utilized for artifact rejection.”*

Q57: - Line 609-611 – are these 1mm/2mm distances to the center of the structures or the border?

A57: The 1 mm/2 mm distances refer to the distance between the center of each contact and the closest voxel within the target. We have clarified the text as follows: *“The BNST and hypothalamus contacts (n = 17) were all within or less than 1 mm from the closest voxel within these structures.”*

Q58: - Line 620 – can report the exact spectral resolution

A58: Thank you for the suggestion. We have revised the text accordingly and specified the spectral resolution as 0.977 Hz.

Q59: - Line 622 – 1/f is a hot topic in the world of biomarkers, including depression (Veerakumar et al, 2019; PMID: 31314668). Did the authors consider this or aperiodic offset as a potential candidate marker in addition to spectral power?

A59: Thank you for this insightful suggestion. We recognize that 1/f activity is a critical component in electrophysiological research and a promising candidate for biomarkers. It is often linked to the passive filtering properties of the extracellular environment and the excitation-inhibition balance in neural networks.

In the early stages of our analysis, we explored the predictive value of the aperiodic component using FOOOF fitting but did not find any significant associations. However, we agree that the aperiodic component, including the 1/f slope, holds considerable potential, particularly in tracking the effects of DBS for treatment-resistant depression. Specifically, we observed some preliminary differences in the 1/f slope between sham and active stimulation during the randomized crossover phase, and we anticipate that further analysis with a larger sample size and longitudinal monitoring could yield more significant results.

Given that this aspect may not align with the current focus of our manuscript and could complicate the discussion, we have opted not to include this analysis in the present study. However, we plan to explore the potential of aperiodic components as a biomarker in future research.

Q60: - Line 631 – further details about cross-validation and hyperparameters would be helpful (see main comment above). Additionally, what is meant by “treatment response”? Was this 3mo improvement compared to preop?

A60: Thank you for your valuable feedback. We have added further details about the cross-validation process and hyperparameters, as well as clarified the definition of “treatment response.” To address your query, “treatment response” refers to the 3-

month improvement compared to preoperative measures. These revisions should provide greater clarity and context.

Q61: - Line 636 – “non-directional” would keep consistent with “Directional” used next for GC analysis

A61: Thank you for your suggestion. We have revised the text to use “non-directional” to keep consistency with “Directional” used in the subsequent GC analysis.

Q62: - Line 641 – report the exact spectral resolution

A62: Thank you for the suggestion. We have revised the text accordingly and specified the spectral resolution as 0.977 Hz.

Q63: - Line 659 – how many participants were missing assessments? And was data taken from time points before or after the missing time point? Unclear how this is directly relevant to the analysis vs clinical trial results so some clarification would be helpful

A63: Thanks for your comments. We had revised the text as follows for clarification:

“For analysis of the clinical data, two patients (P8, P12) missed assessments at months 3 and three patients (P16, P18, P22) at months 12; we used data from the previous visit the month prior in the analysis.”

Q64: - Line 668 – what is meant by “stepwise method” here? It appears that linear regressions were performed with single variable

A64: Thank you for the comment. We used a stepwise regression approach to explore whether physiological theta or neuropsychological biomarkers could predict improvements in quality of life or disability. Through predictor selection, we found that the neuropsychological biomarker, negative emotional bias, rather than theta biomarkers, predicted disability outcomes as measured by SDS improvements at 3 months, 6 months, and 1 year. The original text was as follows:

“We then investigated whether physiological theta or a neuropsychological biomarker could predict improvements in quality-of-life or disability. Using stepwise regression analysis with predictor selection, we found that the neuropsychological biomarker, negative emotional bias, rather than theta biomarkers, predicted disability outcomes measured by SDS improvements at 3 months ($R^2 = 0.45$, $p = 0.003$), 6 months ($R^2 = 0.37$, $p = 0.009$), and 1 year ($R^2 = 0.36$, $p = 0.017$) (Extended Data Fig. 3b).”

Q65: - Line 670 – could add relevant citation(s) for mediation models, especially use in similar analyses

A65: Thank you for the suggestion. We have added relevant citations to support the use of mediation models in the analyses.

Q66: - Line 677 – typo delete the ‘

A66: Thank you for catching that typo. We have removed the unnecessary apostrophe as suggested.

Figures

Q67: - In general, the figure captions could be more descriptive to aid in interpretation, especially for these complex correlation matrices.

A67: Thank you for the valuable suggestion. We have revised the figure captions to provide more detailed descriptions, including clarifications on the variables, axes, and key findings to aid interpretation, particularly for complex correlation matrices. This ensures that readers can better understand and interpret the results independently.

Q68: - The use of “Imp” and “Bs” in figures is not super intuitive. Would be helpful for reader if the authors spelled out variables whenever possible to aid in interpretation

A68: Thank you for the suggestion. We have revised the figures to spell out the variables “Imp” and “Bs” for greater clarity and ease of interpretation by the reader.

Q69: - Fig 1: nice overview. Should figure read “N=15 eyes open” to match text? Typo in electroencephalography in caption.

A69: Thank you for your careful review. We have updated the figure to read “N=15 eyes open” to match the text and corrected the typo in electroencephalography in the caption.

Q70: - Fig 2: the rationale for this large correlation matrix and its relevance to the present study is unclear. The takeaway from this figure is rather unclear. If the authors do keep it, I would recommend pulling key correlations for this main figure to

demonstrate the main point and instead having a supplementary table with R/p-values for reader to reference if interested

A70: We appreciate the reviewer's insightful feedback on Figure 2. In response to both your suggestion and that of Reviewer 2, we have moved this figure to the Extended Data to improve the clarity and focus of the main text. Additionally, we have emphasized key correlations by highlighting relevant data points with black square marks to enhance interpretability.

Q71: - Fig 3: in panel a, the axes should be labeled with improvement. Also, could clarify in caption, neurophysiological signals are from perioperative off-stim, correct?

A71: Revised as suggested. We have clarified that the neurophysiological signals are from the perioperative baseline.

Q72: - Fig. 4: a and b in the figure caption should be switched. Panel c is quite the complex figure. More description in the caption would be helpful to know what is the main takeaway readers should be getting from this figure. Are p-values not corrected?

A72: Thank you for the detailed feedback. We have switched panels a and b in the figure caption as suggested. Additionally, we expanded the description of panel c to better highlight the main takeaway for readers. We also clarified that the neurophysiological signals are from the perioperative baseline. Regarding the p-values, we have added a note to indicate if they are corrected.

Q73: - Fig. 5: Panel a – I am curious about the differences observed between sham/active vs active/sham groups. It appears all patients in sham/active group show the expected relationship between theta and HAMD, compared to only 2/5 in active/ sham. What do the authors hypothesize? Could be an interesting discussion point. Additionally, it could be clearer to display all patients with consistent ordering of stages (i.e., post-open label, post-active, post-sham) to aid in easy comparison. In panel b why is the post-open label (stage 1) not included in the group analysis? Also caption for panel b should say various frequency bands instead of just theta power (as all are presented).

A73: Thank you for raising these insightful points.

Regarding the differences observed between the sham/active vs. active/sham groups, we hypothesize that the divergent relationship between theta and HAMD observed after active stimulation in the sham/active group or after open-label chronic stimulation in the active/sham group, may reflect individual variability in the prolonged effects of stimulation. To note, with our DBS device, wireless recording can only be conducted off-stimulation, and stimulation artifacts were difficult to fully

control, which limits our ability to accurately capture the effects of stimulation. While the prolonged effects of stimulation may persist after being turned off, their impact may vary across subjects. We acknowledge this as a limitation in our study, and it has been addressed in the manuscript:

“Moreover, wireless recordings were conducted off-stimulation, and stimulation artifacts were difficult to adequately control, limiting our ability to accurately capture stimulation effects and potentially explaining the divergent relationship between theta and HAMD observed after active or chronic stimulation.”

We appreciate the reviewer’s suggestion for panel a. In response, we have modified the ordering of stages to ensure consistency across all patients, facilitating easier comparison. The revised figure now presents stages in the uniform sequence of Post-Open Label, Post-Active, and Post-Sham, improving clarity and interpretability.

For panel b, the post-open label stage was excluded from the group analysis because it lasted much longer (around 1 year) than both the post-sham (two weeks) and post-active (two weeks) stages. Thus, it was not directly comparable, and we elected not to include in this analysis.

Finally, thank you for your suggestion regarding the caption for panel b. We have updated it to reflect the use of various frequency bands, rather than focusing solely on theta power:

“b. BNST power across different frequency bands from post sham stimulation (2 weeks) stage to post active stimulation (2 weeks) stage”.

Reviewer #1 (Remarks on code availability):

The code appears to include basic processing of the LFP and EEG data and the ridge regression. It does not include a README. I did not attempt to install or run the code.

Reviewer #2 (Remarks to the Author):

The manuscript by Wang et al. reports biomarker findings from a randomized crossover clinical trial of BNST-NAc DBS for treatment-resistant depression. BNST-NAc DBS is reported to decrease symptoms as measured using a variety of scales. Using perioperative LFP and EEG recordings, the authors demonstrate that BNST theta power and prefrontal-BNST theta coherence predicted the outcome as measured by different rating scales. Using recordings off the DBS device, they demonstrate that BNST theta tracked depression severity longitudinally over the course of the randomized crossover phase of the trial. Additionally, they demonstrate that BNST theta power is correlated with negative emotional arousal bias. The study is timely, given the increasing interest in device-based treatments for treatment-resistant depression and the scarcity of neurophysiological measurements from targeted regions during treatment. The crossover trial design is an effective way to disentangle placebo effects, and the results from BNST-NAc-VIC neurophysiology is an advance beyond existing studies. Overall, the findings are consistent, indicating that lower theta activity is associated with symptom improvement, possibly driven by changes in anxiety. However, despite the potential impact, it is difficult to be enthusiastic about the current manuscript given the significant number of important details and methodological concerns that are currently lacking. These issues, which I will detail below, would need to be addressed before I could assess the potential impact of this manuscript and make any recommendation about publication.

Q1: Given the emphasis on randomized crossover clinical trial, clinical outcomes have not been described in sufficient detail. It is not clear how many of the participants responded to treatment or how many reached remissions. In addition, the results from the crossover phase of the study have not been detailed at all. Therefore, it is not clear if DBS effects were lost during the sham stimulation arm of the study. Ideally, the depression severity measures during the sham and active weeks should be compared and included in the results.

A1: Thank you for your insightful comments. We understand the importance of detailing clinical outcomes, and we apologize for the lack of clarity in this area. Unfortunately, we are unable to report the full data from the randomized crossover clinical trial as it has already been submitted in another journal. However, we have added the following information to the manuscript for clarity:

“Based on the HAMD scores at last follow-up in the open label phase, 13 of 26 patients (50%) were classified as responders, with 9 of them (35%) achieving remission, while the remaining 13 patients (50%) were classified as non-responders, with 7 of them (27%) being partial responders.”

Q2: It is unclear why the authors chose a 2-day washout period for the crossover phase. Since treatment-resistant depression is generally a slowly changing disorder, as compared to Parkinson' disease, a 2-day washout period might not be sufficient.

On a related note, the BNST-theta power remained low in the sham phase for 3/5 patients who started the crossover study with stim, suggesting a carryover effect.

A2: Thank you for your valuable feedback. We have addressed your concern by adding a limitation regarding the 2-day washout period. As noted in the study by Bergfeld et al. (2016), future crossover studies should consider phases of no longer than one week to ensure patient safety, with an adequate washout period to minimize possible carryover effects.

We chose the 2-day washout period based on considerations of patient tolerance and feasibility, taking into account the clinical practice and pilot studies. In our cohort, some patients demonstrated a carryover effect, as indicated by low BNST-theta power during the sham phase, suggesting the need for longer washout periods in future studies to better mitigate these effects. We have now clarified this point in the manuscript, as follows:

“The potential compound effects of stimulation on neural activity and neuroplasticity, coupled with the absence of a defined washout period, raise the possibility of carryover effects. Notably, BNST-theta power remained low in the sham phase for some patients who started the crossover study with stimulation, suggesting a potential carryover effect and highlighting the need for longer washout periods in future studies to mitigate these effects.”

Q3: The details of LFP processing are unclear. Did the authors average the spectra/connectivity measures calculated from individual recordings or the timeseries? (Line 611 – 613) Averaging timeseries seems inappropriate, as this process destroys important temporal information. Also, it is not clear why the authors averaged the left and right hemispheres and the 7 EEG channels, as this process removes spatial information that might be relevant.

A3: Thank you for the insightful comment. In this study, we averaged timeseries across hemispheres before calculating spectra and connectivity measures. Our approach aligns with prior studies for resting state LFP data aimed at increasing the signal-to-noise ratio, minimizing channel selection bias, and facilitating individual-level analyses. Examples include:

- [1] Ricciardi et al., 2021 (Mov Disord, 36(9):2126-2135), who employed similar methods for Parkinson's disease neurophysiology.
- [2] Manssuer et al., 2023 (Brain, 146(6):2642-2653), investigating habenula activity in human subjects.

Since our focus was on resting-state data, we did not prioritize temporal variations, assuming stability over short time windows. Thus, averaging time series from both hemispheres was considered appropriate for calculating spectra and connectivity measures.

Additionally, averaging across EEG channels aimed to avoid spatial selection biases and increase the signal-to-noise ratio given the sparsity of channels. However,

recognizing the potential relevance of spatial specificity, we have also conducted additional analyses to explore whether distinct prefrontal EEG channels exhibit anatomical specificity, guided by tractography findings. The results of these analyses are included in the manuscript.”

“To assess the anatomical specificity of prefrontal EEG channels and their alignment with tractography, we conducted an exploratory analysis between EEG measures (spectral power and EEG-BNST coherence) from each channel and improvements in depression (HAMD, MADRS), anxiety (HAMA), and anhedonia (DARS) at 3 months. Notably, only theta power in FP2 (HAMA: Spearman’s $r = -0.61$, FDR adjusted $p = 0.034$; MADRS: Spearman’s $r = -0.59$, FDR adjusted $p = 0.046$) and F8 (HAMA: Spearman’s $r = -0.66$, FDR adjusted $p = 0.026$; MADRS: Spearman’s $r = -0.59$, FDR adjusted $p = 0.046$) showed significant associations with 3-month HAMA and MADRS improvements. Additionally, theta power across all prefrontal EEG channels was significantly correlated with 3-month HAMD improvements: FP1 (Spearman’s $r = -0.59$, FDR adjusted $p = 0.025$), FP2 (Spearman’s $r = -0.66$, FDR adjusted $p = 0.025$), Fz (Spearman’s $r = -0.56$, FDR adjusted $p = 0.025$), F3 (Spearman’s $r = -0.53$, FDR adjusted $p = 0.03$), F4 (Spearman’s $r = -0.57$, FDR adjusted $p = 0.025$), F7 (Spearman’s $r = -0.55$, FDR adjusted $p = 0.025$), and F8 (Spearman’s $r = -0.62$, FDR adjusted $p = 0.025$)(Extended Data Fig. 1c).”

Interestingly, the associations for FP2 and F8 align with recent clinical interest in targeting these sites for TMS treatment in depression, both of which have demonstrated clinical efficacy and are consistent with the anatomical localization observed in tractography findings.

Q4: Additionally, it is not clear why the authors expect eyes open and eyes closed resting state LFP to be different enough to be considered as separate datasets. While EEG studies often use this distinction for alpha band activity that is clearly modulated with eyes open/eyes closed, it is not clear if the same differences are observed in BNST, NAc, or VIC LFP.

A4: Thank you for your insightful comment. We acknowledge that considering eyes-open and eyes-closed conditions as separate datasets may be inappropriate. Instead, we view them as distinct sensory-attentional states and a replicated within-subject data set to evaluate the robustness of the physiological marker as a predictive biomarker. This distinction allows us to explore potential variations in network activity that might arise from different sensory contexts. While alpha band modulations are well-documented for eyes-open and eyes-closed EEG recordings, we agree that it remains unclear whether similar effects are present in subcortical regions such as BNST, NAc, or VIC. Our study seeks to answer this question by evaluating the generalizability of findings across these states, as clarified in our updated text: “confirmed across eyes-open and -closed states and machine learning” and “Using the eyes-open dataset, we verified generalizability across both states.”

Q5: It is not clear how many samples of the LFP and EEG features per participant were used in the penalized ridge regression model. Similarly, the details of the cross-validation (e.g. leave-n-subject-out, k-fold) have not been reported.

A5: Thank you for raising this point. We apologize for the lack of clarity in our initial submission. To address your concerns, here are the details:

For each participant, we used a single sample of LFP and EEG features from each of the five frequency bands (spectral power: 2 regions × 5 bands; coherence: 5 bands), resulting in a total of 15 features per participant.

Regarding cross-validation, we employed a built-in 3-fold cross-validation for model hyperparameter optimization. Specifically, we split the data into three non-overlapping subsets. For each fold, one subset was used for validation, and the remaining two subsets were used for training. This process was repeated three times, ensuring each subset was used for validation once. Additionally, the model was trained on the 3-month improvement data and applied to the 6-month and 12-month outcomes, with R^2 scores reported for model performance.

We hope this clarification helps, and we have updated the manuscript to include these details for greater transparency.

Q6: Given that EEG and LFP may have differences in signal-to-noise ratio, the inference of directionality from Granger causality may be biased. The authors could implement time-reversed GC to confirm the directionality. See for example: Haufe S, Nikulin VV, Müller KR, Nolte G. A critical assessment of connectivity measures for EEG data: a simulation study. *Neuroimage*. 2013 Jan 1;64:120-33. doi: 10.1016/j.neuroimage.2012.09.036. Epub 2012 Sep 21. PMID: 23006806.

A6: Thank you for this insightful methodological suggestion. Although our analysis focused on correlations rather than paired comparisons between frontal → BNST and BNST → frontal directionality, and is therefore less prone to biases in Granger causality inference, potential confounding remains a consideration.

To address this, we have addressed this point by computing Granger causality at the theta band for time-reversed signals and comparing it to standard non-reversed results using a one-sided paired t-test. Our analysis revealed that the standard theta-frequency Granger causality difference (frontal → BNST minus BNST → frontal) was significantly higher than the time-reversed difference, with positive values in 14 of 17 subjects compared to only 7 of 17 for the time-reversed condition ($t(16) = 1.894$, $p = 0.038$). Additionally, reversed correlations were observed in bottom-up theta band Granger causality in the time-reversed data.

We have incorporated this method into the manuscript with the following statement:

“Time reversal of the time series confirmed the directionality, with reversed correlations observed in bottom-up (HAMA: Spearman’s $r = -0.28$, $p = 0.268$; MADRS: Spearman’s $r = -$

0.57, $p = 0.018$; HAMD: Spearman's $r = -0.48$, $p = 0.05$; uncorrected) but not in top-down theta band Granger causality”;

“Granger causality can be sensitive to differences in signal-to-noise ratios between time series, potentially leading to spurious directionality. To address this, we computed Granger causality for time-reversed signals and compared it to the non-reversed standard results⁶⁶. Time reversal was expected to reverse the direction of true causality while preserving spurious effects due to confounding between-area power differences or signal-to-noise.”

Q7: Given that the authors demonstrate the BNST-theta marker is strongly associated with anxiety (using VAS) rather than mood, it would be useful to see if the prediction of 3-month and 6-month HAM-D and MADRS scores are driven by predictions of changes in anxiety-related symptoms rather than other symptoms like depressed mood, sadness or lassitude

A7: Thank you for raising this critical point. To assess whether BNST-theta predictions of 3- and 6-month HAM-D/MADRS scores were driven by anxiety-related symptoms, we conducted the following analyses:

First, we calculated R^2 values from linear regression models predicting improvements in anxiety (HAMA) and mood-related scales (HAMD, MADRS) at 3, 6, and 12 months. BNST-theta exhibited comparable predictive power for HAMD and HAMA (e.g., 3-month HAMD $R^2 = 0.55$, HAMA $R^2 = 0.54$, MADRS $R^2 = 0.32$).

Partial Least Squares (PLS) regression further confirmed similar predictive power for HAMD/MADRS and HAMA at 3 months (loadings: -0.73 for HAMA vs. -0.74 for HAMD vs. -0.57 for MADRS) and 6 months (loadings: -0.59 for HAMA vs. -0.57 for HAMD vs. -0.51 for MADRS), with a slightly stronger association for HAMA at 1 year (loadings: -0.78 for HAMA vs. -0.69 for HAMD vs. -0.57 for MADRS). However, the strong correlation between HAMA and HAMD improvements ($r = 0.81$, $p < 0.001$) limited the ability to fully dissociate their predictive effects.

Notably, while HAMD-17 captures a multidimensional symptom profile (including somatic and anxiety components), MADRS primarily evaluates sadness and lassitude and is considered unifactorial. The relatively lower predictive value of MADRS compared to HAMA and HAMD suggest that predictions of HAMD and MADRS scores might be driven by changes in anxiety-related symptoms.

To further explore whether BNST-theta's predictive power for HAMD was driven by anxiety-related symptoms, on an exploratory level we examined its association with HAMD subscales: (1) Depressive Obstruction: HAMD items 1, 2, 8, 17; (2) Sleep Disturbance: HAMD items 4, 5, 6; (3) Anxiety: HAMD items 9, 10, 11, 14; (4) Somatic Disorders: HAMD items 5, 11, 13, 14; (5) Depressive Agitation: HAMD items 4, 9, 12, 16. Baseline BNST-theta was significantly correlated with improvements in sleep disturbance ($r = -0.52$, $p = 0.031$), anxiety ($r = -0.51$, $p = 0.036$), somatic symptoms ($r = -0.50$, $p = 0.040$), and depressive agitation ($r = -0.50$, $p = 0.043$) but not depressive obstruction ($p > 0.05$) at 3 months.

The dissociation between anxiety and mood was more evident at shorter timescales (e.g., acute VAS ratings) than in aggregated clinical scales. At 3 and 6 months, symptom overlap in HAMD/MADRS likely diluted specificity, whereas VAS ratings may have captured transient anxiety states with greater dissociation from mood symptoms. We discuss in the manuscript that this dissociation in BNST physiometers observed in VAS scores may reflect heterogeneity in symptomatology and pathophysiology across timescales, with transient anxiety states contributing greater variability and context dependence. See the revised text in line 407-411:

“While depression and anxiety severity are often highly correlated in MDD patients, the dissociation between anxiety and mood states in BNST physiometers in this study may reflect heterogeneous symptomatology and physiopathology across timescales, with transient states introducing greater variability and context dependence⁵⁰.”

Q8: As a suggestion for clarity, I would recommend streamlining the results section around the paper's main message regarding the neurophysiological findings. Results that don't directly address this, such as correlations between symptom severity scores (Fig 2b), could be moved to the supplement.

A8: Thank you for the helpful suggestion. We have streamlined the results section to better emphasize the paper's core neurophysiological findings. In line with your recommendation, we have moved results that do not directly address the central message, such as correlations between symptom severity scores (Fig. 2b), to the Supplementary Materials.

Reviewer #3 (Remarks to the Author):

References:

1. Johnson, K. A., Okun, M. S., Scangos, K. W., Mayberg, H. S. & de Hemptinne, C. Deep brain stimulation for refractory major depressive disorder: a comprehensive review. *Mol Psychiatry* (2024) doi:10.1038/s41380-023-02394-4.
2. Dougherty, D. D. *et al.* A Randomized Sham-Controlled Trial of Deep Brain Stimulation of the Ventral Capsule/Ventral Striatum for Chronic Treatment-Resistant Depression. *Biol Psychiatry* **78**, 240–248 (2015).
3. Holtzheimer, P. E. *et al.* Subcallosal cingulate deep brain stimulation for treatment-resistant depression: a multisite, randomised, sham-controlled trial. *Lancet Psychiatry* **4**, 839–849 (2017).
4. Bergfeld, I. O. *et al.* Deep Brain Stimulation of the Ventral Anterior Limb of the Internal Capsule for Treatment-Resistant Depression: A Randomized Clinical Trial. *JAMA Psychiatry* **73**, 456–64 (2016).
5. Mayberg, H. S. *et al.* Deep Brain Stimulation for Treatment-Resistant Depression. *Neuron* **45**, 651–660 (2005).
6. Figeo, M. *et al.* Deep Brain Stimulation for Depression. *Neurotherapeutics* **19**, 1229–1245 (2022).
7. Coenen, V. A. *et al.* Superolateral medial forebrain bundle deep brain stimulation in major depression: a gateway trial. *Neuropsychopharmacology* **44**, 1224–1232 (2019).
8. Davidson, B. *et al.* Lack of clinical response to deep brain stimulation of the medial forebrain bundle in depression. *Brain Stimul* **13**, 1268–1270 (2020).
9. Zhang, C. *et al.* Bilateral Habenula deep brain stimulation for treatment-resistant depression: clinical findings and electrophysiological features. *Transl Psychiatry* **12**, 52 (2022).
10. Bewernick, B. H. *et al.* Deep brain stimulation to the medial forebrain bundle for depression- long-term outcomes and a novel data analysis strategy. *Brain Stimul* **10**, 664–671 (2017).
11. Bewernick, B. H. *et al.* Nucleus Accumbens Deep Brain Stimulation Decreases Ratings of Depression and Anxiety in Treatment-Resistant Depression. *Biol Psychiatry* **67**, 110–116 (2010).
12. Lozano, A. M. *et al.* Subcallosal Cingulate Gyrus Deep Brain Stimulation for Treatment-Resistant Depression. *Biol Psychiatry* **64**, 461–467 (2008).
13. Jennings, J. H. *et al.* Distinct extended amygdala circuits for divergent motivational states. *Nature* **496**, 224–228 (2013).
14. Marcinkiewicz, C. A. *et al.* Serotonin engages an anxiety and fear-promoting circuit in the extended amygdala. *Nature* **537**, 97–101 (2016).
15. Li, H. *et al.* Neurotensin orchestrates valence assignment in the amygdala. *Nature* **608**, 586–592 (2022).
16. Kim, S. Y. *et al.* Diverging neural pathways assemble a behavioural state from separable features in anxiety. *Nature* **496**, 219–223 (2013).
17. Giardino, W. J. *et al.* Parallel circuits from the bed nuclei of stria terminalis to the lateral hypothalamus drive opposing emotional states. *Nat Neurosci* **21**, 1084–1095 (2018).
18. Yang, B., Karigo, T. & Anderson, D. J. Transformations of neural representations in a social behaviour network. *Nature* **608**, 741–749 (2022).
19. Mei, L., Yan, R., Yin, L., Sullivan, R. M. & Lin, D. Antagonistic circuits mediating infanticide and maternal care in female mice. *Nature* **618**, 1006–1016 (2023).

20. Raymaekers, S., Luyten, L., Bervoets, C., Gabriëls, L. & Nuttin, B. Deep brain stimulation for treatment-resistant major depressive disorder: a comparison of two targets and long-term follow-up. *Transl Psychiatry* **7**, e1251–e1251 (2017).
21. Fitzgerald, P. B. *et al.* A pilot study of bed nucleus of the stria terminalis deep brain stimulation in treatment-resistant depression. *Brain Stimul* **11**, 921–928 (2018).
22. Neumann, W. J. *et al.* Different patterns of local field potentials from limbic DBS targets in patients with major depressive and obsessive compulsive disorder. *Mol Psychiatry* **19**, 1186–1192 (2014).
23. Mosley, P. E. *et al.* A randomised, double-blind, sham-controlled trial of deep brain stimulation of the bed nucleus of the stria terminalis for treatment-resistant obsessive-compulsive disorder. *Transl Psychiatry* **11**, 190 (2021).
24. Meyer, G. M. *et al.* Deep Brain Stimulation for Obsessive-Compulsive Disorder: Optimal Stimulation Sites. *Biol Psychiatry* **96**, 101–113 (2024).
25. Fitzgerald, P. B. *et al.* No Consistent Antidepressant Effects of Deep Brain Stimulation of the Bed Nucleus of the Stria Terminalis. *Brain Sci* **14**, 499 (2024).
26. Baldermann, J. C. *et al.* Connectivity Profile Predictive of Effective Deep Brain Stimulation in Obsessive-Compulsive Disorder. *Biol Psychiatry* **85**, 735–743 (2019).
27. Alagapan, S. *et al.* Cingulate dynamics track depression recovery with deep brain stimulation. *Nature* (2023) doi:10.1038/s41586-023-06541-3.
28. Hacker, C. *et al.* Aperiodic (1/f) Neural Activity Robustly Tracks Symptom Severity Changes in Treatment-Resistant Depression. *Biol Psychiatry Cogn Neurosci Neuroimaging* **10**, 186–194 (2025).
29. Choi, K. S., Riva-Posse, P., Gross, R. E. & Mayberg, H. S. Mapping the “Depression Switch” During Intraoperative Testing of Subcallosal Cingulate Deep Brain Stimulation. *JAMA Neurol* **72**, 1252 (2015).
30. Lee, S., Kim, H., Park, M. J. & Jeon, H. J. Current Advances in Wearable Devices and Their Sensors in Patients With Depression. *Front Psychiatry* **12**, (2021).
31. Jiang, Z. *et al.* Classifying Major Depressive Disorder and Response to Deep Brain Stimulation Over Time by Analyzing Facial Expressions. *IEEE Trans Biomed Eng* **68**, 664–672 (2021).
32. Riva-Posse, P. *et al.* Defining Critical White Matter Pathways Mediating Successful Subcallosal Cingulate Deep Brain Stimulation for Treatment-Resistant Depression. *Biol Psychiatry* **76**, 963–969 (2014).
33. Riva-Posse, P. *et al.* A connectomic approach for subcallosal cingulate deep brain stimulation surgery: prospective targeting in treatment-resistant depression. *Mol Psychiatry* **23**, 843–849 (2018).
34. Scangos, K. W., Makhoul, G. S., Sugrue, L. P., Chang, E. F. & Krystal, A. D. State-dependent responses to intracranial brain stimulation in a patient with depression. *Nat Med* **27**, 229–231 (2021).
35. Scangos, K. W. *et al.* Closed-loop neuromodulation in an individual with treatment-resistant depression. *Nat Med* **27**, 1696–1700 (2021).
36. Mandali, A. *et al.* Acute Time-Locked Alpha Frequency Subthalamic Stimulation Reduces Negative Emotional Bias in Parkinson’s Disease. *Biol Psychiatry Cogn Neurosci Neuroimaging* **6**, 568–578 (2021).
37. Sonkusare, S. *et al.* Power signatures of habenular neuronal signals in patients with bipolar or unipolar depressive disorders correlate with their disease severity. *Transl Psychiatry* **12**, 72 (2022).
38. Avery, S. N., Clauss, J. A. & Blackford, J. U. The Human BNST: Functional Role in Anxiety and Addiction. *Neuropsychopharmacology* **41**, 126–141 (2016).

39. Blomstedt, P., Naesström, M. & Bodlund, O. Deep brain stimulation in the bed nucleus of the stria terminalis and medial forebrain bundle in a patient with major depressive disorder and anorexia nervosa. *Clin Case Rep* **5**, 679–684 (2017).
40. Figeo, M. & Mayberg, H. The future of personalized brain stimulation. *Nat Med* **27**, 196–197 (2021).
41. Solomon, E. A. *et al.* Theta-burst stimulation entrains frequency-specific oscillatory responses. *Brain Stimul* **14**, 1271–1284 (2021).
42. Rao, V. R. *et al.* Direct Electrical Stimulation of Lateral Orbitofrontal Cortex Acutely Improves Mood in Individuals with Symptoms of Depression. *Current Biology* **28**, 3893-3902.e4 (2018).
43. Feffer, K. *et al.* 1 Hz rTMS of the right orbitofrontal cortex for major depression: Safety, tolerability and clinical outcomes. *European Neuropsychopharmacology* **28**, 109–117 (2018).
44. Cui, H. *et al.* A novel dual-site OFC-dlPFC accelerated repetitive transcranial magnetic stimulation for depression: a pilot randomized controlled study. *Psychol Med* **54**, 3849–3862 (2024).
45. Klooster, D. C. W. *et al.* Technical aspects of neurostimulation: Focus on equipment, electric field modeling, and stimulation protocols. *Neurosci Biobehav Rev* **65**, 113–141 (2016).
46. Hollunder, B. *et al.* Mapping dysfunctional circuits in the frontal cortex using deep brain stimulation. *Nat Neurosci* **27**, 573–586 (2024).
47. Rajamani, N. *et al.* Deep brain stimulation of symptom-specific networks in Parkinson’s disease. *Nat Commun* **15**, 4662 (2024).
48. Li, N. *et al.* A unified connectomic target for deep brain stimulation in obsessive-compulsive disorder. *Nat Commun* **11**, 3364 (2020).
49. Faravelli, C., Albanesi, G. & Poli, E. Assessment of depression: A comparison of rating scales. *J Affect Disord* **11**, 245–253 (1986).
50. Saviola, F. *et al.* Trait and state anxiety are mapped differently in the human brain. *Sci Rep* **10**, 11112 (2020).
51. Zheng, Y. *et al.* Validity and Reliability of the Chinese Hamilton Depression Rating Scale. *British Journal of Psychiatry* **152**, 660–664 (1988).
52. Uher, R. *et al.* Measuring depression: comparison and integration of three scales in the GENDEP study. *Psychol Med* **38**, 289–300 (2008).
53. Carmody, T. J. *et al.* The Montgomery Åsberg and the Hamilton ratings of depression: A comparison of measures. *European Neuropsychopharmacology* **16**, 601–611 (2006).
54. Montgomery, S. A. & Åsberg, M. A New Depression Scale Designed to be Sensitive to Change. *British Journal of Psychiatry* **134**, 382–389 (1979).
55. Lai, Y. *et al.* Structural and functional correlates of the response to deep brain stimulation at ventral capsule/ventral striatum region for treatment-resistant depression. *J Neurol Neurosurg Psychiatry* **94**, 379–388 (2023).
56. Li, N. *et al.* A Unified Functional Network Target for Deep Brain Stimulation in Obsessive-Compulsive Disorder. *Biol Psychiatry* **90**, 701–713 (2021).
57. Etkin, A., Egner, T. & Kalisch, R. Emotional processing in anterior cingulate and medial prefrontal cortex. *Trends Cogn Sci* **15**, 85–93 (2011).
58. Stein, D. J. *et al.* Obsessive-compulsive disorder. *Nat Rev Dis Primers* **5**, 52 (2019).
59. Disner, S. G., Beevers, C. G., Haigh, E. A. P. & Beck, A. T. Neural mechanisms of the cognitive model of depression. *Nat Rev Neurosci* **12**, 467–477 (2011).
60. Steiner, A. R. W., Petkus, A. J., Nguyen, H. & Loebach Wetherell, J. Information processing bias and pharmacotherapy outcome in older adults with generalized anxiety disorder. *J Anxiety Disord* **27**, 592–597 (2013).

61. Harmer, C. J., Goodwin, G. M. & Cowen, P. J. Why do antidepressants take so long to work? A cognitive neuropsychological model of antidepressant drug action. *British Journal of Psychiatry* **195**, 102–108 (2009).
62. Sheline, Y. I. *et al.* Increased amygdala response to masked emotional faces in depressed subjects resolves with antidepressant treatment: an fMRI study. *Biol Psychiatry* **50**, 651–658 (2001).
63. Godlewska, B. R., Browning, M., Norbury, R., Cowen, P. J. & Harmer, C. J. Early changes in emotional processing as a marker of clinical response to SSRI treatment in depression. *Transl Psychiatry* **6**, e957–e957 (2016).
64. Kube, T. Biased belief updating in depression. *Clin Psychol Rev* **103**, 102298 (2023).
65. Page, C. E., Epperson, C. N., Novick, A. M., Duffy, K. A. & Thompson, S. M. Beyond the serotonin deficit hypothesis: communicating a neuroplasticity framework of major depressive disorder. *Mol Psychiatry* **29**, 3802–3813 (2024).
66. Haufe, S., Nikulin, V. V., Müller, K.-R. & Nolte, G. A critical assessment of connectivity measures for EEG data: A simulation study. *Neuroimage* **64**, 120–133 (2013).

Response to Reviewers v2 for

Predicting Therapeutic Depression Outcomes: Intracranial Recordings of the Prefrontal – Bed Nucleus of the Stria Terminal Network

Linbin Wang^{1,2,3†}, Yingying Zhang^{2†}, Yuhan Wang^{1†}, Qiong Ding³, Luling Dai¹, Kejia Hu¹, Kuanghao Ye¹, Xin Lv¹, Xiaoxiao Zhang¹, Alekhya Mandali³, Luis Manssuer³, Saurabh Sonkusare³, Yijie Zhao², Peng Huang¹, Xian Qiu¹, Yixin Pan¹, Yijie Lai¹, Dianyou Li¹, Wei Liu¹, Shikun Zhan¹, Bomin Sun^{1‡}, Valerie Voon^{1,2,3‡}

Corresponding author: Valerie Voon, Email: vv247@cam.ac.uk and Bomin Sun, Email: sbm11224@rjh.com.cn

REVIEWER COMMENTS

Reviewer #1 (Remarks to the Author):

The authors have sufficiently addressed all of my comments.

Reviewer #1 (Remarks on code availability):

The code appears to include basic processing of the LFP and EEG data and the ridge regression. It does not include a README. I did not attempt to install or run the code.

We thank Reviewer #1 and #3 for their positive evaluation and confirmation that our revisions have addressed all comments.

Reviewer #2 (Remarks to the Author):

The authors have addressed many comments satisfactorily. However, these concerns still remain:

Q1: The k-fold cross-validation procedure described will lead to data leakage, i.e., the model is exposed to patient-specific information. The choice of this cross-validation raises questions about generalizability across patients. The authors will need to demonstrate similar results with leave-n-subject-out cross-validation or remove references to cross-validation and generalizability across the manuscript and include a discussion of this limitation. Also, the comparison of eyes open and eyes closed data suggests that the biomarker is robust to recording conditions/states, which is very useful information. However, the main question about biomarkers being generalizable is whether they can predict across participants. So, using the term generalizable in the context of eyes open and closed is misleading.

A1: Thank you for this important comment. We fully acknowledge the concern regarding potential data leakage when using k-fold cross-validation across samples from the same subjects. We attempted to implement a leave-n-subject-out cross-validation (LNSO-CV) approach to better assess generalizability across participants. However, due to the limited number of subjects in our dataset, applying LNSO-CV led to folds with insufficient or even negative numbers of training samples, making the analysis statistically unreliable and unstable.

To address this limitation, we have taken the following steps: First, we clarified in the revised manuscript that the current cross-validation approach evaluates within-subject performance rather than across-subject generalizability. Second, we removed references to generalizability across participants throughout the manuscript. Finally, we added a dedicated section discussing this limitation and outlining the need for larger datasets in future studies to properly evaluate cross-subject prediction performance, as follows:

“Fifth, the limited sample size prevented rigorous cross-subject validation, and k-fold cross-validation may have led to data leakage. Therefore, generalizability across participants remains unproven. Future studies with larger cohorts and leave-subject-out validation are needed.”

We also agree with the reviewer’s point that demonstrating robustness to recording states (e.g., eyes open vs. eyes closed) is informative but should not be interpreted as generalizability across individuals. We have revised the manuscript accordingly to avoid potential misinterpretation as follows: *“Using the eyes-open data set, we verified the biomarker is robust across different recording states.”*

Q2: While the authors provide the rationale for averaging LFP timeseries across the hemispheres, the approach still raises concerns for the following reasons. Averaging timeseries across hemispheres destroys temporal dynamics through constructive and destructive interference. It is recommended to average in the spectral domain to preserve temporal dynamics. Furthermore, there may be hemispheric differences in the regions of interest, given that the correlations with HAMA and MADRS are right-dominant. Therefore, the authors are recommended to re-run analyses without averaging across hemispheres.

A2: We thank the reviewer for the valuable feedback regarding the averaging of LFP timeseries across hemispheres. We agree that addressing this issue enhances the depth and rigor of the study. Accordingly, we have added a dedicated results section analyzing left and right hemispheres separately, accompanied by an expanded discussion on the clinical relevance and implications of these findings.

Results: Right-Hemisphere Dominance of Theta-Band Biomarkers

“Hemispheric lateralization is consistently observed in emotion processing, with the left hemisphere linked to positive or approach-related stimuli and the right to negative or withdrawal-related stimuli¹. We assessed hemispheric differences in theta biomarkers by analyzing the left (n = 13) and right (n = 14) BNST separately and examining their correlations with 3-month clinical improvements. Notably, significant correlations with depression and anxiety improvement were observed in the right BNST (HAMA: Spearman’s $r = -0.84$, $p < 0.001$; HAMD: Spearman’s $r = -0.76$, $p = 0.002$; MADRS: Spearman’s $r = -0.71$, $p = 0.004$; DARS: Spearman’s $r = 0.47$, $p = 0.09$), while the left BNST showed no significant associations (HAMA: Spearman’s $r = -0.29$, $p = 0.33$; HAMD: Spearman’s $r = -0.15$, $p = 0.62$; MADRS: Spearman’s $r = -0.33$, $p = 0.27$; DARS: Spearman’s $r = 0.14$, $p = 0.64$).

To assess the anatomical specificity of prefrontal EEG channels and their alignment with tractography, we conducted an exploratory analysis between EEG measures (spectral power and EEG-BNST coherence) from each channel and improvements in depression (HAMD, MADRS), anxiety (HAMA), and anhedonia (DARS) at 3 months and controlled for multiple comparisons using FDR correction across 7 EEG channels. Notably, only theta power in FP2 (HAMA: Spearman’s $r = -0.61$, FDR adjusted $p = 0.034$; MADRS: Spearman’s $r = -0.59$, FDR adjusted $p = 0.046$) and F8 (HAMA: Spearman’s $r = -0.66$, FDR adjusted $p = 0.026$; MADRS: Spearman’s $r = -0.59$, FDR adjusted $p = 0.046$) showed significant associations with 3-month HAMA and MADRS improvements. Additionally, theta power across all prefrontal EEG channels was significantly correlated with 3-month HAMD

improvements: FP1 (Spearman's $r = -0.59$, FDR adjusted $p = 0.025$), FP2 (Spearman's $r = -0.66$, FDR adjusted $p = 0.025$), Fz (Spearman's $r = -0.56$, FDR adjusted $p = 0.025$), F3 (Spearman's $r = -0.53$, FDR adjusted $p = 0.03$), F4 (Spearman's $r = -0.57$, FDR adjusted $p = 0.025$), F7 (Spearman's $r = -0.55$, FDR adjusted $p = 0.025$), and F8 (Spearman's $r = -0.62$, FDR adjusted $p = 0.025$).

Significant associations between EEG–BNST coherence and HAMD improvement were observed exclusively for the right BNST at multiple EEG sites (Fz: Spearman's $r = -0.58$, FDR adjusted $p = 0.043$; F3: Spearman's $r = -0.66$, FDR adjusted $p = 0.043$; F4: Spearman's $r = -0.60$, FDR adjusted $p = 0.043$; F7: Spearman's $r = -0.60$, FDR adjusted $p = 0.043$; F8: Spearman's $r = -0.59$, FDR adjusted $p = 0.043$). Although coherence with the right BNST also correlated with MADRS and HAMA improvements, these did not survive correction for multiple comparisons. No significant associations were observed for the left BNST (Extended Data Fig. 5c)."

Discussion:

"The lateralization of theta biomarkers is clinically meaningful and may reflect circuit-level abnormalities in depression. Although the precise mechanisms remain unclear, lateralized emotion regulation likely plays a role. Importantly, these findings highlight the therapeutic potential of neuromodulation targeting the right prefrontal–BNST circuit or unilateral BNST deep brain stimulation for depression. This is supported by a prior proof-of-concept case demonstrating that closed-loop stimulation of the right ventral capsule/ventral striatum (VC/VS), coupled with sensing at the right amygdala, effectively alleviated depressive symptoms²."

Reviewer #3 (Remarks to the Author):

Reference:

1. Palomero-Gallagher, N. & Amunts, K. A short review on emotion processing: a lateralized network of neuronal networks. *Brain Struct Funct* 227, 673–684 (2022).
2. Scangos, K. W. *et al.* Closed-loop neuromodulation in an individual with treatment-resistant depression. *Nat Med* 27, 1696–1700 (2021).